# Known Drugs Identified by Structure-Based Virtual Screening Are Able to Bind Sigma-1 Receptor and Increase Growth of Huntington Disease Patient-Derived Cells

**DOI:** 10.3390/ijms22031293

**Published:** 2021-01-28

**Authors:** Theo Battista, Gianmarco Pascarella, David Sasah Staid, Gianni Colotti, Jessica Rosati, Annarita Fiorillo, Alessia Casamassa, Angelo Luigi Vescovi, Barbara Giabbai, Marta Stefania Semrau, Sergio Fanelli, Paola Storici, Ferdinando Squitieri, Veronica Morea, Andrea Ilari

**Affiliations:** 1Institute of Molecular Biology and Pathology, National Research Council of Italy, 00185 Rome, Italy; theo.battista@uniroma1.it (T.B.); pascarella.1890179@studenti.uniroma1.it (G.P.); davidsasah.staid@uniroma1.it (D.S.S.); gianni.colotti@cnr.it (G.C.); 2Department of Biochemical Sciences “A. Rossi Fanelli”, “Sapienza” University, 00185 Rome, Italy; annarita.fiorillo@uniroma1.it; 3Cellular Reprogramming Unit, Fondazione IRCCS Casa Sollievo della Sofferenza, 71013 San Giovanni Rotondo, Italy; j.rosati@css-mendel.it (J.R.); a.casamassa@css-mendel.it (A.C.); vescovia@gmail.com (A.L.V.); 4Department of Environmental, Biological and Pharmaceutical Sciences and Technologies, University of Campania Luigi Vanvitelli, 81100 Caserta, Italy; 5Protein Facility, Structural Biology Lab, Elettra Sincrotrone Trieste, 34149 Basovizza, Italy; Barbara.giabbai@elettra.eu (B.G.); marta.semrau@elettra.eu (M.S.S.); paola.storici@elettra.eu (P.S.); 6Department of Cellular, Computational and Integrative Biology—CIBIO, University of Trento, 38123 Trento, Italy; 7Huntington and Rare Diseases Unit, Fondazione IRCCS Casa Sollievo della Sofferenza, 71013 San Giovanni Rotondo, Italy; sergiofanelli.bio@gmail.com (S.F.); f.squitieri@css-mendel.it (F.S.)

**Keywords:** Huntington disease (HD), sigma-1 receptor (σ1R), drug repositioning, virtual screening, computational docking, structure analysis, surface plasmon resonance (SPR), cellular models

## Abstract

Huntington disease (HD) is a devastating and presently untreatable neurodegenerative disease characterized by progressively disabling motor and mental manifestations. The sigma-1 receptor (σ1R) is a protein expressed in the central nervous system, whose 3D structure has been recently determined by X-ray crystallography and whose agonists have been shown to have neuroprotective activity in neurodegenerative diseases. To identify therapeutic agents against HD, we have implemented a drug repositioning strategy consisting of: (i) Prediction of the ability of the FDA-approved drugs publicly available through the ZINC database to interact with σ1R by virtual screening, followed by computational docking and visual examination of the 20 highest scoring drugs; and (ii) Assessment of the ability of the six drugs selected by computational analyses to directly bind purified σ1R in vitro by Surface Plasmon Resonance and improve the growth of fibroblasts obtained from HD patients, which is significantly impaired with respect to control cells. All six of the selected drugs proved able to directly bind purified σ1R in vitro and improve the growth of HD cells from both or one HD patient. These results support the validity of the drug repositioning procedure implemented herein for the identification of new therapeutic tools against HD.

## 1. Introduction

The sigma-1 receptor (σ1R) is an intracellular receptor expressed in central nervous system regions and known to regulate calcium signaling and cell protection. The σ1R gene encodes a 24 kDa protein of 223 amino acids, which is anchored to the endoplasmic reticulum and plasma membranes [1]. The σ1R sequence is not homologous to other mammalian proteins. The closest σ1R homolog is the yeast Δ8-Δ7 sterol isomerase ERG2p enzyme. However, no enzymatic activity has been demonstrated for σ1R to date [2]. σ1R is known to interact with several proteins and, as a consequence, to be involved in many physiological functions, including inter-organelle signaling [3,4,5,6,7,8,9].

The endogenous ligand of this receptor is still unknown, although N,N-dimethyl tryptamine, sphingosine, and myristic acid have been proposed to act as σ1R endogenous modulators [10]. Many molecules have been shown to have agonist or antagonist activity, based on their ability to recapitulate the phenotype of receptor overexpression or knockdown, respectively [11]. Agonists have been associated with cytoprotective activity [12,13]. Antagonists have analgesic effect in both animals and humans [14,15,16]. Additionally, σ1R antagonists potentiate signaling by G-protein coupled receptors [17,18] whereas agonists increase IP3-dependent calcium flux, at least with IP3 receptors type 3 [19], and inhibit sodium and potassium channel current [20]. However, little is known about the molecular mechanisms underlying agonist and antagonist activity. The two ligand classes have been proposed to have a different effect on the oligomeric state of the protein, which would be increased by antagonists and decreased by agonists [21].

The first determination of the three-dimensional (3D) X-ray structure of human σ1R, bound to two different ligands, shed light on protein architecture and allowed the identification of a hydrophobic cavity where both ligands bind [22] (Figure 1). σ1R structure comprises a single transmembrane region, located at the N-terminus (residues 8 to 32) and a C-terminal globular domain (residues 33–223) with the ligand-binding site at its center. In the crystal, the receptor exists in a trimeric quaternary arrangement (Figure 1). The three protomers are intimately associated to form a flat triangle, with a transmembrane region at each corner. The overall fold of the protein is different from those of other proteins crystallized to date, but the core of the C-terminal domain contains a cupin-like β-barrel that encloses the bound ligands. The cupin fold is conserved in a wide variety of proteins, many of which are bacterial metalloenzymes; for this reason, an enzymatic activity of σ1R was hypothesized [2].

The ligands bound to the hydrophobic pocket are named PD144418 (Figure 1 and Appendix A), which has antagonist activity, and 4-IBP, whose activity has not been clearly classified. Both molecules interact with the highly conserved Glu172 (Appendix A). A second essential acidic residue, Asp126, forms a 2.7 Å hydrogen bond with Glu172. The other residues lining the pocket are all hydrophobic. In particular, Val84, Trp89, Met93, Leu95, Leu105, Phe107, Ile124, Trp164 and Leu182 interact with hydrophobic regions of the bound ligands, whereas Tyr103 engages both ligands in an aromatic stacking interaction. In spite of providing a detailed description of σ1R-ligand interactions, the X-ray structure of these complexes did not allow the mechanism of σ1R activation to be understood.

Recently, the X-ray crystal structure of σ1R in complex with two additional antagonists, namely haloperidol and NE-100, and the (+)-pentazocine agonist, has also been determined [2]. Intriguingly, the overall structure of σ1R did not change significantly upon agonist vs. antagonist binding. The major difference consists in a 1.8 Å shift between helices α4 and α5 in the (+)-pentazocine bound structure with respect to the antagonists bound structures. In agreement with the experimentally demonstrated ability of agonists to diminish the degree of σ1R polymerization, molecular dynamics simulations suggested that agonist binding causes a more pronounced shift of the α4 helix with respect to the antagonist, resulting in trimer dissociation [2].

While the detailed mechanism of σ1R activity has not been unveiled yet, modulation of σ1R activity by agonists has been shown to significantly attenuate oxidative stress, neuroinflammation, hypoxia, apoptotic pathways, and other processes caused by neurodegenerative diseases such as Huntington disease (HD) [23], Alzheimer’s disease (AD) [24], Parkinson’s Disease (PD) [25] and amyotrophic lateral sclerosis (ALS) [26]. Therefore, many attempts have been made to design efficient σ1R agonists in the hope that they would be able to counteract the pathogenicity of such diseases. A number of approaches used to reach this goal encompassed the generation of pharmacophore models for σ1R:(i)A first 2D-pharmacophore model, named Glennon–Ph, was designed in the early 90′s in the absence of structural information on the protein [27]. It was based on the structural features of a series of diphenylalkylamine σ1R ligands, and consists in one positively ionisable group (i.e., a basic amine group) and two opposite hydrophobic regions at 2.5–3.9 Å and 6–10 Å from the amine group, respectively, without any angle constraint.(ii)A second pharmacophore model was designed based on the alignment of PD144418, spipethiane, haloperidol and (+)-pentazocine [28]. This model consists in one aromatic region, one nitrogen atom that acts as hydrogen bond acceptor, and another polar feature representing one oxygen or sulphur atom.(iii)A 3D-pharmacophore model was developed based upon 23 structurally diverse molecules with K_i_ values for σ1R between 10 pM and 100 μM [29]. This consists in one positively ionisable group and four hydrophobic features. The model is in good agreement with the first one [27] and lacks the secondary polar binding region of the second [28].(iv)Subsequent models share the presence of one positively ionisable group and several hydrophobic features, with variations in distances and angles, and include the presence of one additional polar group [30,31,32].(v)Two new σ1R pharmacophore models were developed, by taking advantage of the information and resolution provided by the X-ray crystallographic structure of σ1R. 5HK1-PhA is based on the four most important interactions between σ1R and the PD144418 antagonist, and comprises: the amine group that interacts with Glu172 and Asp126; one hydrophobic feature for the interaction between the propyl chain of the ligand and the protein residues Ile124 and His154; and two additional hydrophobic features for the interactions of the phenyl ring and methyl group of the ligand, respectively, with Leu182, Tyr206, and Ile178. 5HK1-PhB was obtained by manually merging the two 5HK1-PhA hydrophobic features that interact with Leu182, Tyr206 and Ile178 [33].

In this paper, we exploited the available structures of σ1R to identify known drugs potentially able to bind the protein and exert a neuroprotective activity. First, we performed a virtual screening of a library of compounds that have been approved by the FDA for clinical use for their ability to bind the receptor. Hits with the highest in silico binding score were tested for their ability to bind σ1R in vitro by Surface Plasmon Resonance (SPR). This analysis allowed us to identify known drugs able to bind σ1R in the micromolar range. Since neurodegenerative diseases may be the principal target of these drugs, we tested them on fibroblasts obtained from skin biopsies of HD patients and healthy donors. Skin primary fibroblasts of HD patients are used as a model for studying the disease, due to the physiological presence of the expanded polyglutamine stretch in the huntingtin protein. Alterations of mitochondrial bioenergetics, increased oxidative stress, morphology and changes in gene expression profile in skin fibroblasts derived from adult HD patients, as well as a significant reduction in cell proliferation, have been described in several studies [34,35,36,37,38]. Since the selected compounds are already approved drugs, they may be both directly evaluated for their clinical efficacy and used as leads for the development of new agents against neurodegenerative diseases.

## 2. Results

### 2.1. Identification of Potential σ1R Binding Drugs by Computational Methods

The 3D structures of σ1R experimentally determined by X-ray crystallography and available from the Protein Data Bank (PDB: https://www.rcsb.org/) [39] are reported in Table 1. All of these structures contain quaternary assemblies of three monomers, each comprising an N-terminal transmembrane helix and C-terminal ligand binding domain. Structure comparisons indicate that the ligand binding domain, which comprises the whole ligand binding site, is highly conserved in all of the monomers present in the different structures, as previously reported by the authors [2,22] and shown by the low root mean square deviation (RMSD) values reported in Appendix A, which are a measure of structural difference based on the distance between equivalent pairs of atoms, in Å. Therefore, the monomer chosen for virtual screening, i.e., chain A in coordinate files 5HK1 (5HK1_A), which is the 3D structure determined with the highest resolution (see Table 1), adequately represents the structures of other monomers.

The chemical formulas, clinical indication and information on the mechanism of action of the 20 virtual screening hits having the best predicted interaction energy with 5HK1_A are reported in Figure 2. The interaction of these 20 FDA drugs with σ1R was further investigated by computational docking to 5HK1_A binding site. This step was performed because the free binding energy predicted by docking methods has an accuracy of ~2–3 kcal/mol standard deviation [40], therefore ranking of poses based on this parameter alone is not reliable. Conversely, highly populated clusters have been shown to be enriched in compounds that show strong binding in experimental tests [41].

The poses of the 20 FDA drugs having the best energy among those in the largest clusters produced by molecular docking, were visually analyzed using the Chimera program [42]. To evaluate the likelihood of ligand-receptor binding, we took into account: i. Parameters calculated by Chimera, such as the number of hydrogen bonds and non-polar interactions, and number of unfavourable van der Waals contacts, if present; and ii. Whether ligand moieties comprised in previously reported pharmacophore models (i.e., the positively charged group and hydrophobic regions) were involved among the aforementioned interactions. Additionally, we visually inspected each of the poses to assess whether additional interactions might occur, in case small conformational adjustments with respect to the poses predicted by docking programs were allowed. As a result of these analyses, six compounds were selected to be experimentally evaluated for σ1R binding by SPR, namely: nilotinib, paliperidone, iloperidone, linagliptin, flibanserin and vilazodone. The complexes of these ligands and of pridopidine, for comparison purposes, with 5HK1_A predicted by molecular docking are shown in Appendix A.

Interestingly, known σ1R ligands were found at much lower virtual screening ranking positions. Pridopidine was at position 458 and haloperidol and N,N-dimethyltriptamine, the highest and lowest ranking among known ligands, were at positions 51 and 1002, respectively. However, the differences in predicted binding energies between the highest ranking known ligands and the 20 highest ranking drugs are relatively small (Table 2a,b), especially when the lack of accuracy of binding energy differences <3 kcal/mol is taken into account [41]. As an example, the binding energy difference between haloperidol and the 1st ranked hit (risperidone) is 1.7 kcal/mol, between haliperidol and pridopidine is 2.2 kcal/mol, and between haloperidol and N,N-dimethyltriptamine is 1.6 kcal/mol. Examination of the poses of the six selected drugs in comparison with that of pridopidine shows that the latter is somewhat smaller and, therefore, establishes a lower number of hydrophobic interactions, while the polar interactions with Glu172 is maintained. The lower number of interactions might account for the worse interaction energy and, therefore, ranking, predicted by Vina.

### 2.2. σ1R Expression and Purification

σ1R protein was expressed and purified as described in the Section 4. As shown in Figure 3, in elution fraction E2 and E3 the protein displays a degree of purity higher than 98%. The protein has been concentrated for SPR experiments to a final concentration of 1 mg/mL.

### 2.3. Assessment of Direct Drug Binding to σ1R In Vitro by SPR

In vitro SPR experiments (Figure 4) show that all the compounds predicted by virtual screening to interact with the receptor are indeed able to bind to σ1R, with dissociation constant (K_D_) values in the micromolar range (Table 3). Interestingly, flibanserin, iloperidone and linagliptin (K_D_ < 10 μM) showed higher affinity values for σ1R than pridopidine (K_D_ about 15 μM). This value falls in a different concentration range with respect to the previously reported inhibition constant value (K_i_ = 81.7 nM) [43], which, however, was determined in significantly different experimental conditions. In the previous assay σ1R was inserted in cell membranes and saturated with [3H](+)-pentazocine; the pridopidine K_i_ value was then measured based on its ability to displace the radioactive ligand. Conversely, the K_D_ value measured in SPR experiments indicates the binding affinity between pridopidine and the purified σ1R in conditions where ligand and receptor are allowed to interact directly.

### 2.4. Pridopidine Effect on Healthy and HD Cell Growth

We compared two healthy fibroblast lines (CTR1 and CTR2) with two HD lines (HD1 and HD2) carrying 43 CAG repeats each, whose donors showed the same initial HD stage. As shown in Figure 5 and Table 4, the number of HD fibroblasts was significantly lower than that of controls after 72 h from plating. The growth rates of HD fibroblasts were also significantly lower than those of controls (Table 5). These data are in agreement with the previously reported alteration in proliferation rate of adult onset HD fibroblasts compared with their healthy counterparts [36]. Additionally, both HD fibroblast lines showed a significantly higher percentage of dead cells with respect to healthy CTR lines (Figure 5b and Table 6).

To investigate whether cell proliferation rate was a suitable parameter to indicate the ability of selected drugs to interact with σ1R in a cellular context and exert an agonist or antagonist function, we cultured fibroblasts in the presence of the known σ1R agonist pridopidine. In the presence of pridopidine, the cell number of both HD and healthy lines was significantly increased with respect to the DMSO control after 72 h and, in the case of HD2 cells, at 48 h as well (Figure 6a–d). This increase was not paralleled by a reduced number of dead cells, which was only reduced to a small, not significant extent (Figure 6e–h).

### 2.5. Effects of Drugs on Healthy and HD Cells Growth

Since pridopidine binding to σ1R positively affects cell proliferation, we investigated the effect of the six selected drugs in the same assay. We treated fibroblast lines with each of the drugs and measured the number of cells (alive and dead) at 48 h and 72 h from plating. All drugs were tested at the same concentration used for pridopidine (i.e., 1 μM), and all data were normalized with respect to the same cell lines treated only with the DMSO vehicle (Figure 7 and Table 7).

All of the selected drugs significant increased fibroblast number at 72 h, in either HD1 or HD2 cell lines, and some of them at 48 h as well. Iloperidone, paliperidone and nilotinib had a consistent beneficial effect, since they significantly increased both HD1 and HD2 fibroblast number at 72 h, and paliperidone at 48 h as well (Figure 7c,d and Table 7). However, while iloperidone did not have any effect on healthy fibroblasts at either 72 or 48 h, paliperidone significantly decreased CTR1 and increased CTR2 cell number at 72 h and nilotinib significantly decreased CTR2 cell number at 72 h (Figure 7a,b and Table 7). The effect of the other drugs on the different cell lines is less consistent. The number of HD1 fibroblasts is significantly increased by linagliptin, flibanserin and vilazodone at 72 h, and by vilazodone at 48 h as well, but none of these drugs increases the number of HD2 fibroblasts, flibanserin even determining a significant reduction in cell number at 72 h (Figure 7c,d and Table 7). As far as control cell lines are concerned, all of these compounds determined significant increases in CTR2 and/or CTR1 cell number: CTR2 cells were increased by linagliptin and vilazodone at 72 h and by linagliptin and flibanserin at 48 h; CTR1 cells were increased by vilazodone at 72 h (Figure 7a,b and Table 7).

Importantly, the effect of the tested drugs on HD fibroblasts was comparable to, and, in several cases, better than that of pridopidine. In particular, at 72 h, the number of HD1 fibroblasts in the presence of linagliptin, paliperidone or vilazodone, and the number of HD2 fibroblasts in the presence of iloperidone or paliperidone, was significantly higher than the number of the respective fibroblasts in the presence of pridopidine (Table 7).

In agreement with the above results, treatment with the selected drugs significantly increased the growth rates of either HD1 or HD2 fibroblasts to an extent comparable to, and, in several cases, better than that of pridopidine (Table 8).

### 2.6. Effects of Drugs on Healthy and HD Cell Death

An increase in cell number in response to drug treatment might correlate with a reduction of cell death. Indeed, three of the selected drugs, namely linagliptin, paliperidone and vilazodone, determined a significant reduction in the number of HD1 dead fibroblasts at both 48 and 72 h. However, none of the compounds decreased significantly HD2 dead cells, and flibanserin even significantly increased their number. Conversely, the presence of selected drugs did not increase the number of dead CTR cells, with the exception of nilotinib, which was toxic for CTR2, and the possible exception of iloperidone, which was toxic for CTR1 at 48 h, but not 72 h (Figure 7e–h and Table 9).

## 3. Discussion

Neurodegenerative diseases, such as HD, PD, AD and ALS, are characterized by progressive brain cell dysfunction and loss of neuron structure and function; this, in turn, causes movement disorders (i.e., chorea, dystonia, parkisonisms) and/or mental (i.e., cognitive decline and behavioural changes). Since neuron loss is an unrepairable process, these diseases are still incurable and represent a heavy burden both for patients and their families, and for the society as a whole. Therefore, there is an urgent need to identify therapeutic agents to treat neurodegenerative diseases at their first stages, when neurodysfunction occurs before massive cell death has taken place, to at least attenuate disease symptoms. However, drug development is notoriously a long, expensive, and high-risk enterprise. Conversely, drug repositioning or repurposing, i.e., the identification of novel applications for drugs that have been developed for different medical applications, offers a higher success vs. risk ratio, as well as requiring considerably less time.

This is due to the fact that these drugs and especially those that have been approved for clinical use by regulatory agencies, have already overcome many of the required steps to reach that stage, ranging from the determination of safety and pharmacokinetic profiles, to the optimization of chemical, manufacturing and formulation features [44].

In the framework of neurodegenerative diseases, σ1R is drawing attention as a therapeutic target because its agonists have been shown to be able to counteract neurodegenerative disease processes. As an example, one of these agonists, pridopidine, is safe [45,46] and has beneficial effects on functional capacity [47] and brain connectivity [48] of HD patients, as well as beneficial effects for HD cell and mouse models [23,49,50,51].

Based on the above, in this work we have exploited computational and experimental methods in a synergistic manner, to identify FDA approved drugs that are able to bind σ1R and ameliorate the pathological HD process in a human cellular model.

We employed a computational strategy consisting of three main steps. First, virtual screening of a library comprising all the FDA-approved drugs publicly available from the ZINC database, as well as known σ1R ligands as controls, provided us with a prediction of the ability of each compound to interact with the σ1R ligand binding site. Second, computational docking of the 20 compounds predicted to bind σ1R with highest affinity generated more accurate models of ligand-receptor complexes and binding affinity predictions. Finally, accurate visual inspection of the complexes with best interaction energy among all the complexes present in the most populated clusters allowed us to check whether unfavourable or favourable interactions were likely to occur, in particular those previously identified in experimentally determined 3D structures of σ1R complexes. This strategy led us to select six FDA-approved drugs, namely iloperidone, paliperidone, flibanserin, linagliptin, vilazodone and nilotinib, as likely σ1R binders.

In vitro SPR experiments demonstrated that the computational strategy was successful, since each of the selected compounds was proved to be able to bind σ1R at concentrations in the micromolar range. Importantly, the value of ligand-σ1R complexes K_D_ is in the same range as that measured for pridopidine, and even lower for three of these compounds (flibanserin, iloperidone, linagliptin). The difference between the K_D_ value measured by SPR for the direct interaction between pridopidine and purified σ1R and the previously published value is likely due to the substantial difference in the experimental set-up, since the previously reported K_i_ value refers to pridopidine ability to displace [3H](+)-pentazocine bound to a membrane preparation of σ1R [43].

Importantly, experiments performed on HD fibroblasts showed that the six selected compounds were able to increase the growth of HD patient fibroblasts, albeit to a different extent from one another, indicating that, like pridopidine, they also exert agonistic activity upon σ1R binding. For cell experiments, we used skin fibroblasts derived from two HD patients in the first phase of their disease and compared their responses to the drugs with the responses of two fibroblast lines obtained from two healthy donors. The choice of fibroblasts is due to the fact that, although HD is a neurodegenerative disease, cellular homeostasis alterations have been observed in peripheral tissues as well [52]. We initially demonstrated that the pridopidine agonist significantly increased cell growth in both HD and healthy fibroblasts. This effect was due to a significant increase in cell number with respect to untreated cells, whereas the percentage of dead cells was not significantly affected. In a previous experiment, pridopidine was reported to lower the percentage of HD dead cells with respect to DMSO treatment [50]. However, the neurons used in that experiment were derived from HD cells carrying 82 CAG repeats in HTT, a mutation mirroring a juvenile-onset disease in humans, a HD variant with a particularly severe HD course and a pathological brain pattern different from adulthood disease [53]. Conversely, our experiments were performed in patient cells carrying 43 CAG repeats, a mutation length identical or similar to that occurring in the majority of patients and obtained from subjects at a very initial HD stage, whose neuron loss is presumably relatively limited.

Among the six tested drugs, iloperidone appears to have the most favourable effects. It had 3-fold higher affinity than pridopidine for σ1R in vitro and significantly improved HD fibroblast growth. Iloperidone activity was specific for HD cells, since it did not affect healthy cell growth to a significant extent. Iloperidone is an anti-psychotic drug used for schizophrenia treatment, which is able to bind several dopamine and serotonin receptor isotypes [54]; in particular, it binds dopamine D2 and serotonin 5-HT2A receptors in the caudate nucleus and putamen of the brain [55]. Interestingly, pridopidine also binds D2 and D3 receptors, although pridopidine affinity for σ1R is 100 and 30-fold higher than for D2 and D3, and the occupancy of these receptors in HD patients is much lower than that of σ1R (about ~3% vs. ~90%, respectively) [56].

Like iloperidone, paliperidone is an anti-psychotic drug, able to bind to several dopamine and serotonin receptors. Accordingly, we found that the two compounds had similar effects on HD fibroblasts: like iloperidone, paliperidone significantly improved the growth of fibroblasts derived from both HD patients; additionally, it also decreased the percentage of dead HD fibroblasts from one patient. Paliperidone in vitro affinity for σ1R was 3- and 9-fold lower than that of pridopidine and iloperidone, respectively.

Nilotinib is the third of our six selected drugs, together with iloperidone and paliperidone, which was able to increase the growth of fibroblasts from both HD patients. It is a tyrosine kinase inhibitor used in the treatment of chronic myelogenous leukemia associated with a specific genetic abnormality called Philadelphia chromosome. It has also been investigated in different types of cancers and, interestingly, in the treatment of diseases associated with an accumulation of intracellular proteins, such as PD and AD. The beneficial effect in these diseases in mouse models has been ascribed to nilotinib’s ability to pass the blood brain barrier and stimulate autophagy, leading to the elimination of toxic protein aggregates [57,58]. The results obtained in the present work suggest that binding to, and exerting an agonistic activity on, σ1R may contribute, at least in part, to nilotinib’s protective activity towards PD and AD.

Vilazodone and linagliptin were able to significantly increase the growth of fibroblasts, as well as decrease the percentage of dead cells, from the HD1 patient only. Vilazodone is a serotonin reuptake inhibitor and partial agonist of 5-HT1A receptor used to treat major depressive disorder [59]. Linagliptin is a dipeptidyl peptidase-4 inhibitor that reduces blood sugar by increasing insulin and decreasing glucagon production by the pancreas, used to treat diabetes mellitus type 2.

Flibanserin has the least consistent effect, since it increased the growth of HD1 fibroblasts and decreased that of HD2 patient fibroblasts, to a significant extent. Flibanserin acts as agonist and antagonist of serotonin 5HT1A and 5HT2A receptors, respectively, and has been approved for female hypoactive sexual desire disorder treatment [60].

Interestingly, with the exception of linagliptin [61], the selected drugs are able to cross the blood-brain barrier. Flibanserin, iloperidone, paliperidone and vilazodone exert their therapeutic activity on brain receptors following oral administration or injection, and nilotinib has been shown to cross the blood-brain barrier following injection [57].

Like most drugs, at therapeutic dosages, these compounds present unwanted side-effects, the most common of which have either mild or moderate severity. Only in the case of nilotinib, a serious risk of severe, possibly life-threatening heart complications has been reported; for this reason, this compound carries a black box warning in the US, which is the strongest warning required by the FDA [62].

Additionally, the therapeutic activities for which these drugs are commonly prescribed must be carefully evaluated for their potential beneficial or detrimental effect in HD patients. In particular, the ability of iloperidone, paliperidone, vilazodone and flibanserin to interact with several dopamine and serotonin receptors in the brain, which might, in principle, contribute to their neuroprotective activity, may also interfere with the psychiatric symptoms of HD patients. Indeed, both selective serotonin reuptake inhibitors and atypical antipsychotics are recommended to treat depression and psychosis in these patients, respectively [63]; however, since both antidepressants, like vilazodone, and atypical antipsychotics, like iloperidone and paliperidone, might induce psychiatric reactions and/or extrapyramidal side-effects (e.g., muscle rigidity, parkinsonism, etc.), the effects of both these classes of compounds in HD patients have to be carefully assessed and balanced by specialist neuropsychiatrists [64].

On the whole, drug treatment induced different responses in healthy and HD fibroblasts, even if the latter were obtained from patients at onset of the disease, and in HD cells from different patients, suggesting that different pathways are involved. Further analyses will have to be performed to identify the signalling pathways activated by the analyzed drugs in HD cells.

Additional studies will have to be performed to ascertain whether one or more of the drugs studied in this work can be advantageously used in HD therapy. These include the investigation of drug activity: on a larger panel of HD patient fibroblasts, to assess the broadness of activity spectrum; on cell models such as iPSCs-derived neurospheres and neurons; in the presence of σ1R antagonists, to evaluate the extent to which their protective activity is specifically mediated by the interaction with this receptor; and, most importantly, in animal models of HD, to assess in vivo activity and safety at potentially therapeutic dosages. The K_i_ values of selected drugs and known agonist can also be measured by radioligand displacement assays with membrane bound σ1R, to determine their correlation with those measured by SPR experiments on purified molecules.

Once further studies confirm the promising results reported herein, the best performing drugs will be directly amenable to off-label clinical use and undergo significantly more time- and cost-effective clinical trials than those required for new molecules whose safety for human use is unknown. Indeed, the available information about drugs that are already in the market comprises not only data acquired during the clinical trials that had been performed to gain FDA approval, but also the much larger amount of data gathered during many years of use in the general population.

In addition to new information about the possibility to ‘reposition’ drugs that are currently used for the therapy of different diseases to attenuate HD symptoms, this work provides a proof-of-concept about the possibility to identify new activities for known drugs by selecting a suitable target receptor, such as σ1R, and applying the procedure described above, which combines computational techniques (i.e., virtual screening, docking and visual analyses), with experimental in vitro validation of drug-receptor interactions and assessment of drug activity in HD patient cells.

To increase the currently available set of therapeutic tools against HD and, possibly, other neurodegenerative diseases, we plan to apply the procedure implemented herein to identify additional FDA-approved σ1R agonists with beneficial activity on HD fibroblasts. Indeed, the ability of all of the six compounds studied in this work to increase the growth of fibroblasts from at least one HD patient demonstrates the ability of computational methods to correctly identify σ1R binders; however, there was no correlation between the predicted binding energy listed in Table 2 and actual binding affinities reported in Table 3. As an example, flibanserin and vilazodone bind purified σ1R with highest and lowest affinity, respectively, the vilazodone K_D_ being 10-fold higher than that of flibanserin, but the differences in predicted binding energy between the two compounds are below 3 kcal/mol and, therefore, as discussed above, they are not significant. It is therefore possible that other compounds among those listed in Table 2, or even other FDA-approved compounds whose complexes with σ1R were not visually inspected, are endowed with agonistic activity towards σ1R. This hypothesis is supported by the fact that pridopidine and other known σ1R agonists are ranked between the 51^st^ (haloperidol) and 1002^nd^ (N,N-dimethyltriptamine) position by Vina, suggesting that several drugs with better, and possibly even with worse, predicted binding energy than N,N-dimethyltriptamine, may exert agonistic activity on σ1R.

Currently, pridopidine is undergoing clinical investigation, since a phase 3 clinical trial named PROOF-HD, to finally test its efficacy as disease modifier in patients (using the TFC score as primary end point indicator) is due to start soon. If successful, this trial may represent the starting point of a new era, when a new class of drugs aiming at modifying the course of HD will be used in the clinic. These drugs might be potentially combined with other therapeutic strategies, such as genetic (i.e., antisense drugs) or gene (i.e., intracerebral viral vector miRNA drugs) therapies that are also currently in clinical development (in phase 3 and phase 1 trials, respectively).

Our future objective is to envisage a strategy to select additional virtual screening hits to investigate by computational docking, visual inspection, SPR and ability to increase HD cells growth, with the aim to identify further drugs that may be advantageously repurposed as potential therapeutic agents against HD.

## 4. Materials and Methods

### 4.1. Identification of Potential σ1R Binding Drugs by Computational Methods

#### 4.1.1. Receptor Preparation

The atomic coordinates of all of the σ1R 3D structures that have been experimentally determined by X-ray crystallography were downloaded from the PDB (https://www.rcsb.org/) [39] (see Table 1). Structure visualization and analysis were performed using the programs Swiss-PDBViewer [65], CHIMERA [42] and PyMol [66]. The σ1R 3D structure in coordinate file 5HK1, chain A (5HK1_A) was chosen for molecular docking, since it was solved with highest resolution and is representative of all the other available monomers (see Section 2 and Appendix A). Crystallographic waters and ligand molecules were removed using Chimera. AutoDock Tools (ADT) v. 1.5.6 was used to add hydrogen atoms, merge non-polar hydrogen atoms and automatically assign Gasteiger charges.

#### 4.1.2. Ligand Preparation

We used a set of molecules comprising: (i) the 1576 FDA approved compounds from the ZINC15 database that are available for sale (http://zinc15.docking.org/) [67]; (ii) all ligands present in the available σ1R 3D structures and listed in Table 1; and (iii) ligands known from the literature to bind σ1R and not already included in the previous lists, namely: captodiame; N,N-dimethyltryptamine; phencyclidine; prasterone; and remoxipride (DrugBank identifiers: DB09014; DB01488; DB03575; DB01708; and DB00409, respectively) [68]. The FDA compounds from Zinc were converted to PDB format in two steps: first, the SMILE format was converted to mol2 with MarvinSketch 18.26 (https://chemaxon.com); second, the mol2 format was converted to PDB using Open Babel v. 2.3.1 [69]. All ligands were then converted in pdbqt format using a script from ADT 1.5.6 [70] using the additional parameters: “-A ‘hydrogen_bonds’” to add hydrogens and build bonds among non-bonded atoms; and “-U ‘nphs_lps’” to merge both non-polar hydrogens and lone pairs.

#### 4.1.3. Virtual Screening

The following space searching parameters were adopted: spacing value at 0.375 Å; center on coordinates 12.168, 36.423 and −34.778; and 30 × 24 × 34 grid points. Virtual screening was performed using the program Vina [71] with the following parameters: “--num_modes 100”, which represents the maximum number of binding modes to generate; and “--energy_range 9”, in order to maximize the energy difference between the best binding mode and the worst one. Additionally, all conformations (poses) were kept rather than only those with a Vina score better than a given threshold. All other parameters had default values. After virtual screening, the 20 hits with the best Vina score were extracted using the “vina_screen_get_top.py” script from Autodock Vina tools [71] (see Table 2).

#### 4.1.4. Molecular Docking

The 20 lowest energy results from virtual screening against 5HK1_A were docked again into the same monomer using ADT. The same space searching parameters reported for virtual screening were used, plus: 100 genetic algorithm runs; population size of 150; RMS cluster tolerance of 2 Å. All other parameters were left as default.

#### 4.1.5. Results Analysis

Python scripts were developed to parse Vina and Autodock output files and perform a preliminary analysis of the 20 ligand-5HK1_A complexes having the lowest energy among all of the poses of the most populated clusters generated by ADT (see Table 2). In particular, the pose energy of each ligand was extracted from the Vina pdbqt file; the total number of poses, mean energy and lowest energy of both the best energy cluster and the largest cluster comprising the selected ligands, were extracted from the Autodock DLG file; features of protein-ligand interactions such as hydrogen bonds, number of contacts and number of unfavourable interactions (clashes) were calculated by the structure visualization and analysis program Chimera, following re-building of receptor-ligand complexes. Information on the clinical indication and mechanism of action of each compound were manually obtained from KEGG [72] and DrugBank [68]. Scripts interfacing with Chimera were written in Python version 2.7, required by Chimera. All other scripts were written in Python version 3.6. Pandas libraries were used for the analysis.

#### 4.1.6. Visual Inspection

The 20 complexes automatically analyzed using Chimera (Table 2) were visually inspected using the program PyMol to assess whether the docked compounds possessed the chemical features shared by ligands in complex with σ1R in known 3D structures, namely: (i) one positive or partially positive charge, to bind to the Glu/Asp pair; and (ii) hydrophobic regions complementary to hydrophobic protein regions.

### 4.2. σ1R Expression and Purification

The pFastBAC plasmid encoding for the FLAG tagged σ1R gene was kindly provided by Prof. Andrew Kruse (Harvard Medical School, Boston, MA, USA). The protein was expressed and purified according to previously described conditions [22] at the Protein Facility of Elettra Sincrotrone Trieste (Basovizza, Trieste, Italy). The bacmid was generated by transposition using *E. coli* DH10Bac competent cells (Invitrogen, Waltham, MA, USA). Sf9 insect cells (Expression Systems, Davis, CA, USA) were used for virus preparation and protein expression. Infection was performed at a density of 1 x 106 cells/mL and cultures were grown at 27 °C for 72 h. Cells were harvested by centrifugation, lysed by osmotic shock in 20 mM Hepes pH 7.5, 2 mM MgCl_2_ and 1:100.000 (*v*/*v*) Benzonase (#E1014, Merck, Darmstad, Germany) and then centrifuged for 15′ at 47,800× g. The pellet was resuspended in solubilisation buffer (20 mM Hepes pH 7.5, 250 mM NaCl, 20% *v*/*v* glycerol, 1% *w*/*v* lauryl maltose neopentyl glycol (LMNG; Anatrace, Maumee, OH, USA), and 0.1% (w/v) cholesteryl hemisuccinate (CHS; Steraloids, Newport, RI, USA)) using a glass Dounce homogeniser. After 2 h stirring at 4 °C, samples were centrifuged for 20′ at 47,800xg. Filtered supernatant was then loaded on Anti–DYKDDDK resin (#L00432, GenScript, Nanjing, China), previously equilibrated with 20 mM Hepes pH 7.5, 100 mM NaCl, 0.2% *v*/*v* glycerol, 0.1% *w*/*v* LMNG, 0.01% *w*/*v* CHS. The resin was first washed with the same buffer used for equilibration, and washed again with a second washing buffer (20 mM Hepes pH 7.5, 100 mM NaCl, 0.02% *v*/*v* glycerol, 0.01% LMNG, 0.001% CHS). The protein was eluted with 0.2 mg/mL FLAG peptide (#RP10586-1, GenScript), and dissolved in the second washing step buffer, in three steps: E1, E2, E3. These steps differ in the time of incubation of the resin with the elution buffer (E1: no incubation; E2: 1 h incubation; E3: over-night incubation). σ1R purity was assessed by SDS–Page and Western blot (Figure 3).

### 4.3. Assessment of Direct Drug Binding to σ1R In Vitro by SPR

SPR experiments were carried out using a SensiQ Pioneer system (SensiQ, Oklahoma City, OK, USA), as described in [73], as follows. The sensor chip (COOH5) was chemically activated by a 35 μL injection of a 1:1 mixture of N-ethyl-N′-(3-(diethylaminopropyl) carbodiimide (200 mM) and N-hydroxysuccinimide (50 mM) at a flow rate of 5 μL/min. The σ1R (ligand) was immobilized on activated sensor chips via amine coupling. Immobilization was carried out in 20 mM sodium acetate at pH 4.0; unreacted groups were blocked by injecting 1 M ethanolamine hydrochloride (35 μL). σ1R immobilization level was 2350 RU.

The drugs (analytes) were dissolved in 100% DMSO at 10 mM concentration, diluted in 20 mM Hepes, 150 mM NaCl, 0.005% surfactant P20 buffer (HBSP) to a concentration of 200 μM (2% final DMSO concentration), further diluted in HBSP + 2% DMSO (HBSP2%D) and injected on the sensor chip at constant flow rate (30 μL/min) at the following times and concentrations: (i) Flibanserin and iloperidone: 0–30 s: 312 nM; 30–60 s: 625 nM; 60–90 s: 1.25 μM; 90–120 s: 2.5 μM; 120–150 s: 5 μM; 150–162 s: 10 μM; (ii) all other analytes: 0–30 s: 780 nM; 30–60 s: 1.56 μM; 60–90 s: 3.12 μM; 90–120 s: 6.25 μM; 120–150 s: 12.5 μM; 150–162 s: 25 μM.

The RU increase relative to baseline indicates complex formation; the plateau region represents the steady-state phase of the interaction (RU_eq_); and the decrease in RU after 162 s represents dissociation of analytes from the immobilized σ1R after injection of HBSP2%D buffer. Regeneration procedures are based on two long (2000 s and 500 s) injections of buffer, separated by a brief (5 s) injection of 10 mM NaOH. Sensorgrams analysis was carried out using Qdat (SensiQ, (SensiQ, Oklahoma, OK, USA) ForteBio) and by performing Scatchard analysis as described in [74].

### 4.4. Ethical Approval

The fibroblasts used in this work were derived from skin biopsies of two healthy human subjects and two HD patients, who had signed fully informed consent forms. All procedures performed in studies involving human participants were conducted in accordance with the standards of the Ethics Committee of the Institute “Casa Sollievo della Sofferenza”. Furthermore, specimen collection and medical procedures were in full accord with the Helsinki declaration (WMA Declaration of Helsinki—Ethical Principles for Medical Research Involving Human Subjects).

### 4.5. Skin Biopsy and Fibroblast Isolation

Skin punch biopsies were obtained from two healthy donors and from two HD patients (cell lines names: HD4503 and HD256.05, named HD1 and HD2 in this work) with identical mutation length (43 CAG repeats each in HTT gene) and preserved independence which was clinically rated by the total functional capacity (TFC) scale score (TFC HD4503: 12; TFC HD256.05: 13, i.e., the first of five HD stages [75]). All subjects (CTR1, CTR2, HD1 and HD2) are male, aged 48, 60, 44 and 38 years at collection, respectively.

Biopsies were cut into small fragments and plated on a tissue-culture dish in FBS (Fetal Bovine Serum; Sigma Aldrich, St. Louis, MI, USA) over night at 37 °C in 5% CO2 atmosphere. For the next 20–30 days the fragments were cultured in fibroblast medium (DMEM high glucose, 20% FBS, 2 mM L-glutamine and 1% penicillin–streptomycin; all reagents from Sigma Aldrich) to allow fibroblasts growth. Once fibroblasts were confluent, they were isolated by trypsin dispersion and amplified in the same medium and conditions.

### 4.6. Growth Rate Analysis in Basal Conditions

Approximately 1 × 10^5^ fibroblasts were seeded in *n* = 4 wells with the same concentration per well. Cell viability and growth curves were determined by direct counting of stained and unstained cells after 48 and 72 h. Cell count was performed in blind conditions using the Bürker Chamber. Cell suspension was added with trypan-blue (5 µL each; dilution factor 1:2) and then introduced into the device by capillary action. The number of viable and non-viable (blue stained) cells was obtained by multiplying the average number of cells in one of the nine squares of the grid determined by microscope observation x the volume of the starting well × the dilution factor × 10,000. The count was carried out three times for each well, and the mean cell count was plotted against culture time. Cell growth rate was calculated by dividing the current value by the previous value. The percentage of dead cells was calculated by dividing the trypan-blue positive cells by the total number of cells and multiplying the resulting values by 100.

### 4.7. Growth Rate Analysis after Drug Treatment

Approximately 1 × 10^5^ cells were seeded in triplicate in a 6-well plate with the same concentration per well and directly treated with drugs or vehicle through for three days. Pridopidine, iloperidone, linagliptin, flibanserin, paliperidone, vilazodone and nilotinib were dissolved in DMSO (Sigma-Aldrich) to obtain a stock solution of 0.5 mM, and diluted in the medium to achieve a final solution of 1 μM. Fibroblasts viability and growth curves were determined by direct counting of stained and unstained cells after 48 and 72 h. Cell count was performed in blind conditions using the Bürker Chamber, as described for cells in basal conditions (see above). The mean cell count/well was recorded and cell numbers were plotted against culture time. Cells growth rate was calculated by dividing the current value by the previous value. The percentage of dead cells was calculated by dividing the trypan-blue positive cells by the total number of cells and multiplying the resulting values by 100.

### 4.8. Statistical Analysis

Statistical analyses were performed with GraphPad Prism 8 software (GraphPad Software, San Diego, CA, USA) using Two-way ANOVA, followed by uncorrected Fisher’s LSD for multiple comparisons when required. Data are shown either as mean ± standard deviation (SD) or as box plot, which represents median, 25 and 75% percentiles, and minimum and maximum values. Statistical significance was assumed at *p* < 0.05.

## Figures and Tables

**Figure 1 ijms-22-01293-f001:**
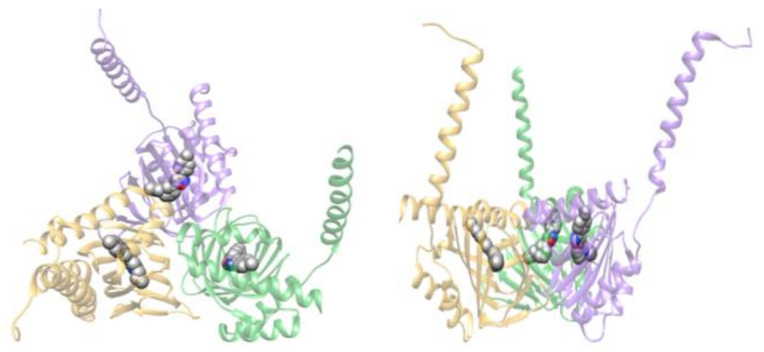
3D structure of σ1R in complex with PD144418 (PDB ID of co-ordinate file: 5HK1). The three identical protein monomers are shown as transparent ribbon and coloured green, yellow and purple, respectively. The ligand is shown as spheres and coloured by atom-type: C, white; N, blue; O, red. Left: top view. Right: side view.

**Figure 2 ijms-22-01293-f002:**
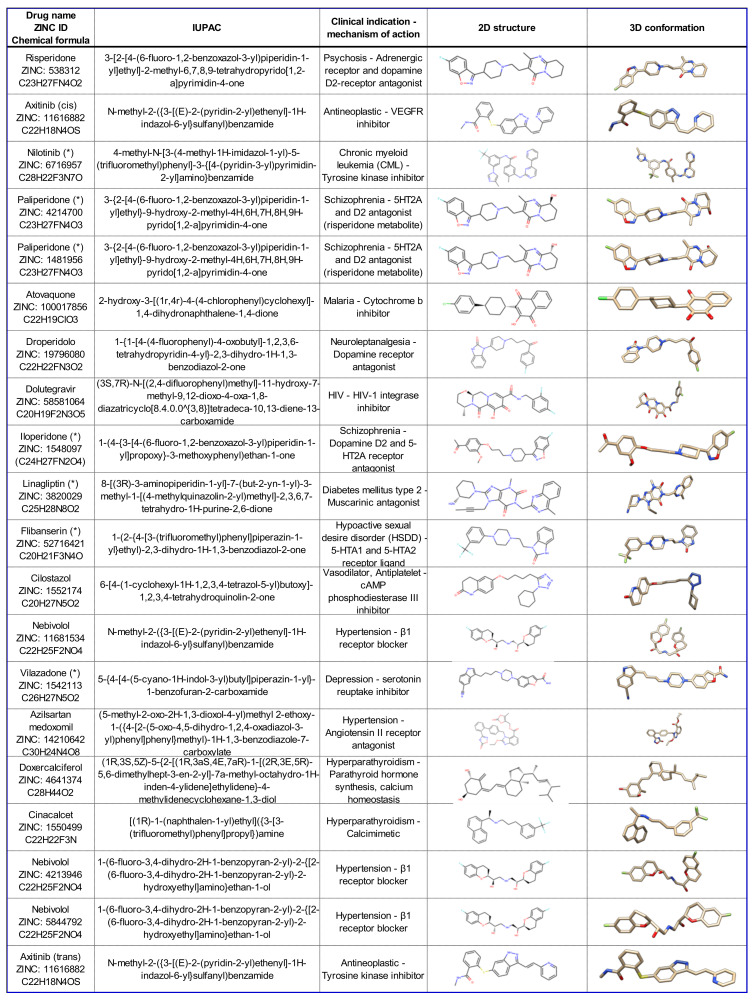
Clinical indication and chemical structures of the compounds listed in Table 2. Compounds selected for experimental validation by SPR are indicated by an asterisk (*).

**Figure 3 ijms-22-01293-f003:**
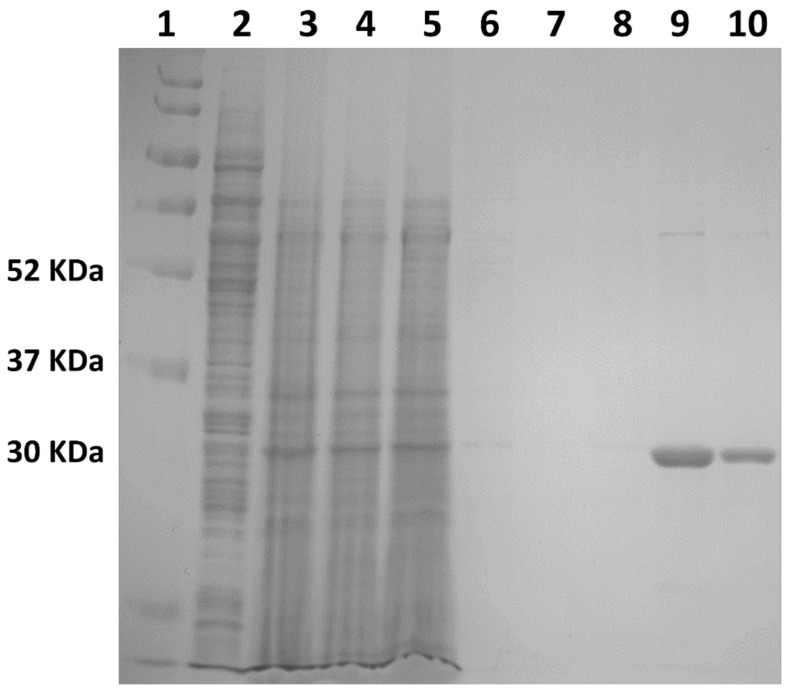
SDS–Page, from left to right: 1, protein ladder; 2, supernatant; 3, pellet; 4, load; 5, flow–through; 6 and 7, wash 1 and 2; 8, 9 and 10, elution fractions E1, E2 and E3, respectively.

**Figure 4 ijms-22-01293-f004:**
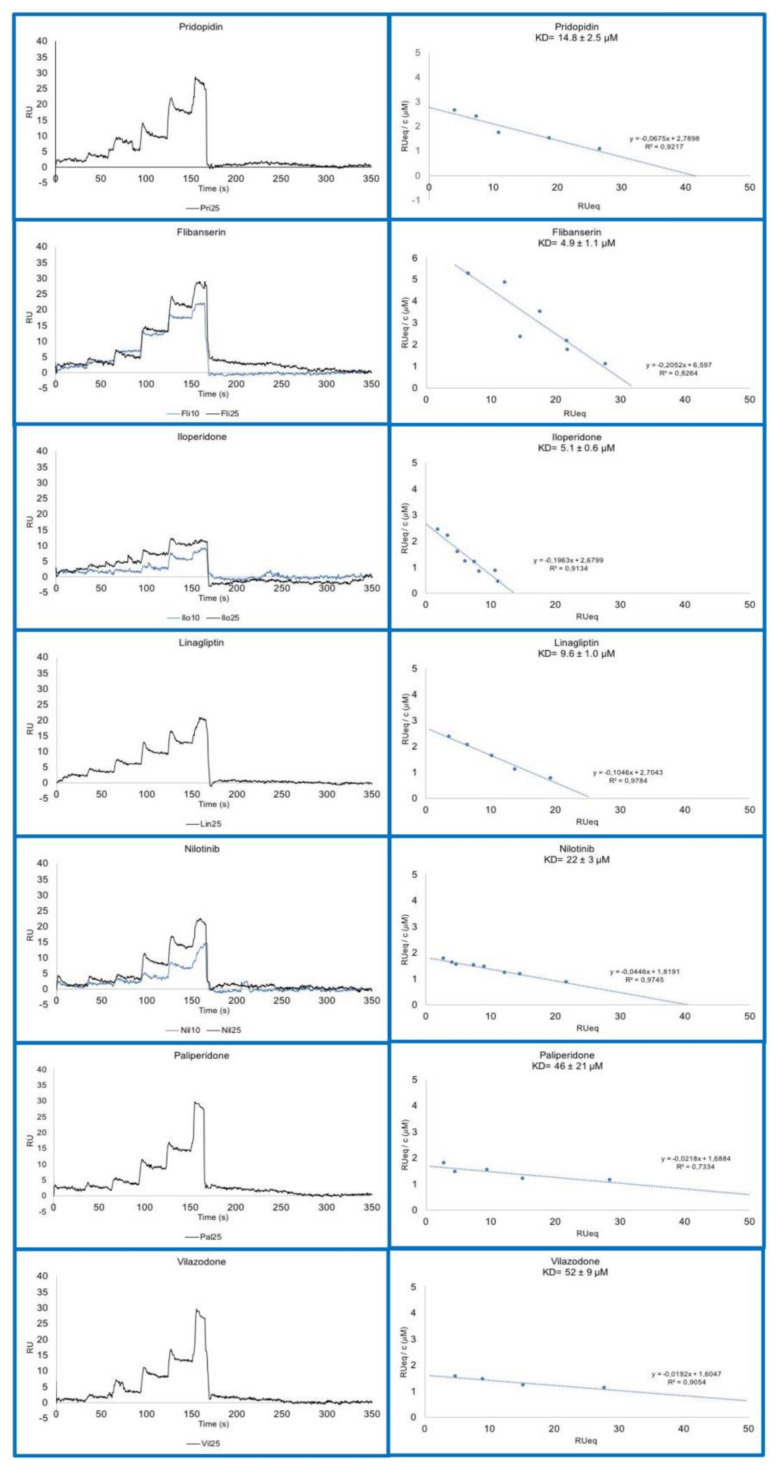
Sensorgrams (**left**) and Scatchard plot analyses (**right**) of compounds selected by virtual screening and computational docking, and of pridopidine as a control.

**Figure 5 ijms-22-01293-f005:**
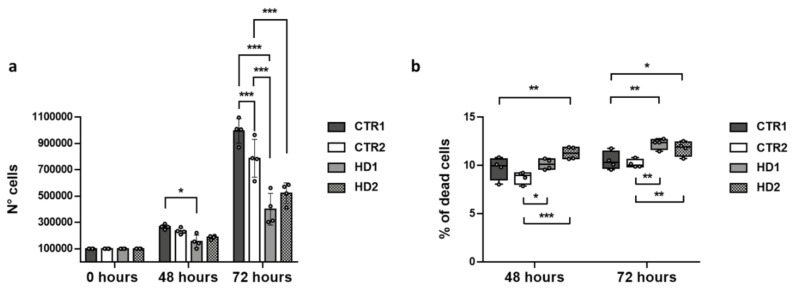
(**a**) Number of CTR1, CTR2, HD1 and HD2 fibroblasts in basal conditions at 0, 48 and 72 h. Data are shown as mean ± SD. (**b**) Percentage of cell death for all cell lines in basal conditions. The box plot represents median, 25 and 75% percentiles, and minimum and maximum values. Fibroblast numbers were determined by direct counting of stained and unstained cells after 48 and 72 h, and plotted against culture time. * *p* < 0.05, ** *p* < 0.01, *** *p* < 0.0001 compared with CTR lines (Two-way ANOVA, followed by uncorrected Fisher’s LSD for multiple comparisons).

**Figure 6 ijms-22-01293-f006:**
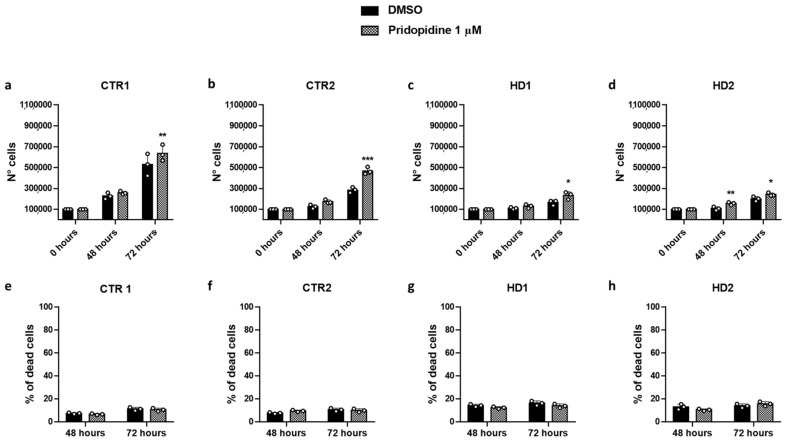
Pridopidine effect on CTR1 (**a**), CTR2 (**b**), HD1 (**c**) and HD2 (**d**) cell growth, and CTR1 (**e**), CTR2 (**f**), HD1 (**g**) and HD2 (**h**) percentage of dead cells. Fibroblast numbers were determined by direct counting of st0ained and unstained cells after 48 and 72 h and plotted against culture time. Data are shown as mean ± SD. * *p* < 0.05, ** *p* < 0.01, *** *p* < 0.0001, compared with DMSO treatment (Two-way ANOVA, followed by uncorrected Fisher’s LSD for multiple comparisons).

**Figure 7 ijms-22-01293-f007:**
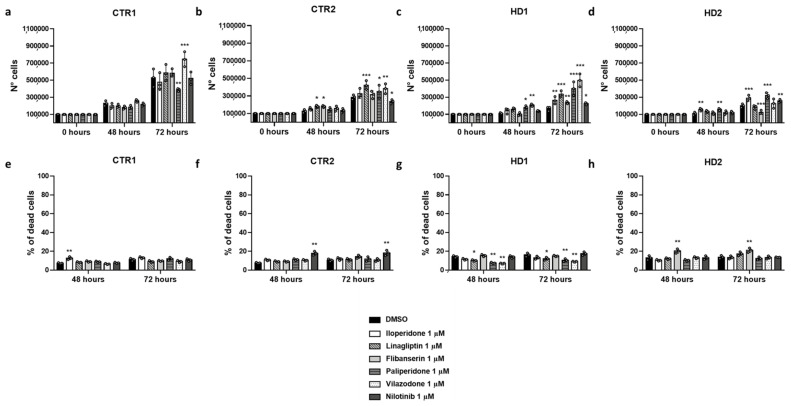
Effect of iloperidone, linagliptin, flibanserin, paliperidone, vilazodone and nilotinib on CTR1 (**a**), CTR2 (**b**), HD1 (**c**) and HD2 (**d**) cell growth, and percentage of dead cells in CTR1 (**e**), CTR2 (**f**), HD1 (**g**) and HD2 (**h**) cell lines. Fibroblast numbers were determined by direct counting of stained and unstained cells after 48 and 72 h and plotted against culture time. Data are shown as mean ± SD. * *p* < 0.05, ** *p* < 0.01, *** *p* < 0.0001, compared with DMSO treatment (Two-way ANOVA, followed by uncorrected Fisher’s LSD for multiple comparisons).

**Table 1 ijms-22-01293-t001:** 3D structures of σ1R experimentally determined by X-ray crystallography and available from the PDB. PDB ID (chain), PDB identifier (name of the chains); Res, Resolution (angstroms); Ligand short (PDB): ligand short name (PDB file name); Ligand long (formula): ligand full name (chemical formula); Ligand function: agonist/antagonist activity towards σ1R; Ref, literature reference.

PDB ID (Chain)	Res (Å)	Ligand Short (PDB)	Ligand Complete (Formula)	Ligand Function	Ref
5HK1 (ABC)	2.51	PD144418 (61W)	3-(4-methylphenyl)-5-(1-propyl-3,6-dihydro-2H-pyridin-5-yl)-1,2-oxazole (C_18_H_22_N_2_O)	Antagonist, nM binder	[22]
5HK2 (ABC)	3.20	4-IBP (61V)	N-(1-benzylpiperidin-4-yl)-4-iodobenzamide (C_19_H_21_IN_2_O)	Antagonist, nM binder	
6DJZ (ABC)	3.08	Haloperidol (GMJ)	4-[4-(4-chlorophenyl)-4-hydroxypiperidin-1-yl]-1-(4-fluorophenyl)butan-1-one (C_21_H_23_ClFNO_2_)	Antagonist	[2]
6DK0 (ABC)	2.90	NE-100 (GKY)	N-{2-[4-methoxy-3-(2-phenylethoxy)phenyl]ethyl}-N-propylpropan-1-amine (C_23_H_33_NO_2_)	Antagonist	
6DK1 (ABC)	3.12	(+)-Pentazocine (GM4)	(2S,6S,11S)-6,11-dimethyl-3-(3-methylbut-2-en-1-yl)-1,2,3,4,5,6-hexahydro-2,6-methano-3-benzazocin-8-ol (C_19_ H_27_NO)	Agonist	

**Table ijms-22-01293-t002a:** (**a**)

Drug Name	ZINC ID Number	Vina Best E (kcal/mol)	Chimera			Vina/ATD Comparison	ATD Largest Cluster			Chimera			ATD Lowest Energy Cluster		
			#hb	#cla	#con	RMSD (Å)	Best E (kcal/mol)	Mean E (kcal/mol)	#pos	#hb	#cla	#con	Best E (kcal/mol)	Mean E (kcal/mol)	#pos
Risperidone	538312	−12.6	0	0	70	2.9	−11.5	−11.1	37	0	1	99	−11.5	−11.1	37
Axitinib (cis)	11616882	−12.6	3	0	59	0.5	−10.9	−10.8	41	1	0	70	−10.9	−10.8	41
Nilotinib (*)	6716957	−12.3	4	1	110	6.4	−7.8	−5.5	16	2	11	148	−9.5	−3.8	6
Paliperidone (*)	4214700	−12.2	0	0	72	3.1	−11.5	−10.7	46	1	2	111	−11.5	−10.9	4
Paliperidone (*)	1481956	−12.2	0	0	80	10.7	−11.1	−10.6	14	1	1	93	−11.9	−11.2	4
Atovaquone	100017856	−11.9	0	0	51	2.9	−11.1	−10.7	73	0	1	68	−11.3	−11.0	27
Droperidolo	19796080	−11.9	0	0	58	3.3	−9.8	−9.4	27	0	1	83	−10.2	−9.9	13
Dolutegravir	58581064	−11.9	3	0	58	9.4	−9.4	−9.2	52	0	0	71	−10.3	−9.9	43
Iloperidone (*)	1548097	−11.7	1	0	70	10.7	−10.2	−9.7	39	0	2	84	−10.4	−9.6	7
Linagliptin (*)	3820029	−11.7	0	1	97	0.6	−12.4	−10.0	33	1	6	112	−12.4	−10.0	33
Flibanserin (*)	52716421	−11.6	0	0	48	1.0	−9.4	−9.2	71	0	0	59	−10.0	−9.4	11
Cilostazol	1552174	−11.6	0	0	71	9.4	−9.2	−8.7	25	1	0	68	−9.7	−9.2	19
Nebivolol	11681534	−11.6	2	0	62	10.1	−10.0	−8.7	73	1	1	81	−10.0	−8.7	73
Vilazodone (*)	1542113	−11.6	2	2	81	6.1	−9.2	−8.2	38	2	7	126	−9.4	−7.6	7
Azilsartan Medoxomil	14210642	−11.5	2	1	123	9.0	−7.9	−4.4	18	0	12	143	−9.0	−4.4	8
Doxercalciferol	4641374	−11.5	0	2	96	3.4	−12.2	−10.1	55	1	2	112	−12.2	−10.1	55
Cinacalcet	1550499	−11.5	0	0	52	8.7	−10.4	−9.8	23	1	0	85	−10.4	−9.8	23
Nebivolol	4213946	−11.5	5	0	62	2.4	−10.9	−9.5	58	2	0	78	−10.9	−9.5	58
Nebivolol	5844792	−11.5	4	0	61	2.5	−11.1	−9.3	62	4	0	84	−11.1	−9.3	62
Axitinib (trans)	11616882	−11	0	0	63	2.3	−7.6	−7.6	75	0	0	61	−7.7	−7.6	15

**Table ijms-22-01293-t002b:** (**b**)

Ligand Name (PDB)	ZINC ID	Vina Best E (kcal/mol)	Ranking
Haloperidol (GMJ)	537822	−10.9	51
4-IBP (61V)	1642602	−10.7	71
PD144418 (61W)	5862	−10.1	139
NE−100 (GKY)	598622	−8.9	406
Pridopidine	22063703	−8.7	458
(+)-Pentazocine (GM4)	596	−8.6	516
Phencyclidine	968311	−8.5	529
Remoxipride	2021799	−7.9	723
Captodiame	2040210	−7.9	733
N,N-Dimethyltryptamine	897457	−7.1	1002

**Table 3 ijms-22-01293-t003:** σ1R affinity for selected compounds and pridopidine, as a control, measured by SPR experiments.

FDA Name	ZINC ID	K_D_ (μM)
Pridopidine	ZINC000022063703	14.8 ± 1.0 (*)
Flibanserin	ZINC000052716421	4.9 ± 1.1
Iloperidone	ZINC000001548097	5.1 ± 0.6
Linagliptin	ZINC000003820029	9.6 ± 1.0
Nilotinib	ZINC000006716957	22.0 ± 3.0
Paliperidone	ZINC000004214700	46.0 ± 21
Vilazodone	ZINC000001542113	52.0 ± 9.0

(*) Pridopidine K_i_ = 81.7 nM in a displacement assay [43].

**Table 4 ijms-22-01293-t004:** Number of CTR1, CTR2, HD1 and HD2 fibroblasts at 48 and 72 h in basal conditions. Number of cells is shown as mean ± standard deviation (SD). * *p* < 0.05 and *** *p* < 0.0001, compared with CTR1 (Two-way ANOVA, followed by uncorrected Fisher’s LSD for multiple comparison); ^###^
*p* < 0.0001, compared with CTR2 (Two-way ANOVA, followed by uncorrected Fisher’s LSD for multiple comparison).

Cell Line	N° Cells 48 h (Mean ± SD)	N° Cells 72 h (Mean ± SD)
CTR1	2.67 × 10^5^ ± 1.81 × 10^4^	9.95 × 10^5^ ± 9.37 × 10^4^
CTR2	2.34 × 10^5^ ± 2.30 × 10^4^	7.86 × 10^5^ ± 1.43 × 10^5^
HD1	1.54 × 10^5^ ± 4.95 × 10^4^ *	4.00 × 10^5^ ± 1.20 × 10^5^ *** ^###^
HD2	1.86 × 10^5^ ± 1.43 × 10^4^	5.22 × 10^5^ ± 7.77 × 10^4^ *** ^###^

**Table 5 ijms-22-01293-t005:** Growth rates of CTR1, CTR2, HD1 and HD2 fibroblasts at 48 and 72 h in basal conditions. Growth rates are shown as mean ± standard deviation (SD). ** *p* < 0.01 and *** *p* < 0.0001, compared with CTR1 (Two-way ANOVA, followed by uncorrected Fisher’s LSD for multiple comparisons); ^#^
*p* < 0.05 and ^##^
*p* < 0.01, compared with CTR2 (Two-way ANOVA, followed by uncorrected Fisher’s LSD for multiple comparisons).

Cell Line	Growth Rate 48 h (Mean ± SD)	Growth Rate 72 h (Mean ± SD)
CTR1	2.67 ± 0.18	3.73 ± 0.11
CTR2	2.34 ± 0.23	3.34 ± 0.30
HD1	1.54 ± 0.50 *** ^##^	2.63 ± 0.34 *** ^##^
HD2	1.86 ± 0.14 ** ^#^	2.79 ± 0.21 *** ^##^

**Table 6 ijms-22-01293-t006:** Percentage of CTR1, CTR2, HD1 and HD2 dead fibroblasts at 48 and 72 h in basal conditions. Percentages of dead cells are shown as mean ± standard deviation (SD). * *p* < 0.05 and ** *p* < 0.01, compared with CTR1 (Two-way ANOVA, followed by uncorrected Fisher’s LSD for multiple comparisons); ^#^
*p* < 0.05, ^##^
*p* < 0.01 and ^###^
*p* < 0.0001, compared with CTR2 (Two-way ANOVA, followed by uncorrected Fisher’s LSD for multiple comparisons).

Cell Line	% Dead Cells 48 h (Mean ± SD)	% Dead Cells 72 h (Mean ± SD)
CTR1	9.71 ± 1.19	10.49 ± 0.93
CTR2	8.75 ± 0.63	10.19 ± 0.43
HD1	10.13 ± 0.56 ^#^	12.25 ± 0.55 ** ^##^
HD2	11.29 ± 0.66 ** ^###^	11.77 ± 0.77 * ^##^

**Table 7 ijms-22-01293-t007:** Growth of CTR1, CTR2, HD1 and HD2 fibroblasts at 48 and 72 h after treatment with DMSO or 1 μM pridopidine, iloperidone, linagliptin, flibanserin, paliperidone, vilazodone or nilotinib. The number of cells is reported as mean ± standard deviation (SD). * *p* < 0.05, ** *p* < 0.01, *** *p* < 0.0001, compared with DMSO treatment (Two-way ANOVA, followed by uncorrected Fisher’s LSD for multiple comparisons); ^#^
*p* < 0.05, ^##^
*p* < 0.01, ^###^
*p* < 0.0001, compared with pridopidine treatment (Two-way ANOVA, followed by uncorrected Fisher’s LSD for multiple comparisons).

Cell Line	Drug Treatment	N° Cells 48 h (Mean ± SD)	N° Cells 72 h (Mean ± SD)
CTR1	DMSO	2.28 × 10^5^ ± 3.01 × 10^4^	5.28 × 10^5^ ± 1.05 × 10^5 ##^
	Pridopidine	2.58 × 10^5^ ± 1.56 × 10^4^	6.37 × 10^5^ ± 7.71 × 10^4^ **
	Flibanserin	1.80 × 10^5^ ± 1.80 × 10^4 #^	5.82 × 10^5^ ± 4.50 × 10^4^
	Iloperidone	1.98 × 10^5^ ± 3.02 × 10^4^	4.78 × 10^5^ ± 9.89 × 10^4 ##^
	Linagliptin	1.98 × 10^5^ ± 1.93 × 10^4^	5.85 × 10^5^ ± 9.45 × 10^4^
	Nilotinib	2.16 × 10^5^ ± 1.80 × 10^4^	5.20 × 10^5^ ± 7.30 × 10^4 ##^
	Paliperidone	1.86 × 10^5^ ± 1.82 × 10^4^	3.89 × 10^5^ ± 1.70 × 10^4^ ** ^###^
	Vilazodone	2.58 × 10^5^ ± 1.60 × 10^4^	7.45 × 10^5^ ± 8.74 × 10^4^ *** ^##^
CTR2	DMSO	1.26 × 10^5^ ± 1.80 × 10^4^	2.86 × 10^5^ ± 2.59 × 10^4 ###^
	Pridopidine	1.74 × 10^5^ ± 1.82 × 10^4^	4.69 × 10^5^ ± 3.44 × 10^4^ ***
	Flibanserin	1.80 × 10^5^ ± 1.11 × 10^4^ *	3.14 × 10^5^ ± 4.26 × 10^4 ###^
	Iloperidone	1.50 × 10^5^ ± 1.57 × 10^4^	3.25 × 10^5^ ± 5.37 × 10^4 ###^
	Linagliptin	1.80 × 10^5^ ± 1.21 × 10^4^ *	4.22 × 10^5^ ± 5.00 × 10^4^ ***
	Nilotinib	1.32 × 10^5^ ± 2.40 × 10^4^	2.35 × 10^5^ ± 2.35 × 10^4^ * ^###^
	Paliperidone	1.44 × 10^5^ ± 2.30 × 10^4^	3.48 × 10^5^ ± 7.39 × 10^4^ * ^###^
	Vilazodone	1.56 × 10^5^ ± 2.80 × 10^4^	3.82 × 10^5^ ± 5.64 × 10^4^ ** ^##^
HD1	DMSO	1.10 × 10^5^ ± 1.00 × 10^4^	1.65 × 10^5^ ± 2.49 × 10^4 #^
	Pridopidine	1.30 × 10^5^ ± 1.86 × 10^4^	2.34 × 10^5^ ± 3.52 × 10^4^ *
	Flibanserin	1.00 × 10^5^ ± 1.92 × 10^4^	2.36 × 10^5^ ± 1.28 x 10^4^ **
	Iloperidone	1.40 × 10^5^ ± 3.21 × 10^4^	2.63 × 10^5^ ± 4.29 × 10^4^ **
	Linagliptin	1.60 × 10^5^ ± 1.92 × 10^4^	3.35 × 10^5^ ± 3.71 × 10^4^ *** ^##^
	Nilotinib	1.38 × 10^5^ ± 8.49 × 10^3^	2.24 × 10^5^ ± 1.76 × 10^4^ *
	Paliperidone	1.80 × 10^5^ ± 2.83 × 10^4^ *	4.00 × 10^5^ ± 1.13 × 10^5^ *** ^###^
	Vilazodone	2.06 × 10^5^ ± 1.98 × 10^4^ ** ^#^	4.97 × 10^5^ ± 1.06 × 10^5^ *** ^###^
HD2	DMSO	1.08 × 10^5^ ± 1.70 × 10^4 ##^	2.01 × 10^5^ ± 2.21 × 10^4 #^
	Pridopidine	1.56 × 10^5^ ± 1.37 × 10^4^ **	2.43 × 10^5^ ± 1.53 × 10^4^ *
	Flibanserin	1.14 × 10^5^ ± 1.60 × 10^4 #^	1.23 × 10^5^ ± 2.16 × 10^4^ *** ^###^
	Iloperidone	1.56 × 10^5^ ± 1.54 × 10^4^ **	2.90 × 10^5^ ± 3.30 × 10^4^ *** ^##^
	Linagliptin	1.32 × 10^5^ ± 1.30 × 10^4^	1.83 × 10^5^ ± 2.46 ×10^4 ##^
	Nilotinib	1.20 × 10^5^ ± 1.55 × 10^4 #^	2.56 × 10^5^ ± 1.92 × 10^4^ **
	Paliperidone	1.56 × 10^5^ ± 1.20 × 10^4^ **	3.19 × 10^5^ ± 3.46 × 10^4^ *** ^###^
	Vilazodone	1.20 × 10^5^ ± 2.20 × 10^4 #^	2.26 × 10^5^ ± 5.46 × 10^4^

**Table 8 ijms-22-01293-t008:** Growth rates of CTR1, CTR2, HD1 and HD2 fibroblasts at 48 and 72 h after treatment with DMSO or 1 μM pridopidine, iloperidone, linagliptin, flibanserin, paliperidone, vilazodone or nilotinib. Growth rates are shown as mean ± standard deviation (SD). * *p* < 0.05, ** *p* < 0.01, *** *p* < 0.0001, compared with DMSO treatment (Two-way ANOVA, followed by uncorrected Fisher’s LSD for multiple comparisons); ^#^
*p* < 0.05, ^##^
*p* < 0.01, ^###^
*p* < 0.0001, compared with pridopidine treatment (Two-way ANOVA, followed by uncorrected Fisher’s LSD for multiple comparisons).

Cell Line	Drug Treatment	Growth Rate 48 h (Mean ± SD)	Growth Rate 72 h (Mean ± SD)
CTR1	DMSO	2.28 ± 0.30	2.30 ± 0.17
	Pridopidine	2.58 ± 0.16	2.46 ± 0.15
	Flibanserin	1.80 ± 0.18 ** ^###^	3.24 ± 0.14 *** ^###^
	Iloperidone	1.98 ± 0.30 ^##^	2.40 ± 0.14
	Linagliptin	1.98 ± 0.19 ^##^	2.94 ± 0.21 ** ^##^
	Nilotinib	2.16 ± 0.18 ^##^	2.40 ± 0.14
	Paliperidone	1.86 ± 0.18 ** ^###^	2.10 ± 0.16 ^#^
	Vilazodone	2.58 ± 0.16	2.88 ± 0.16 ** ^##^
CTR2	DMSO	1.26 ± 0.18 ^##^	2.28 ± 0.12 ^##^
	Pridopidine	1.74 ± 0.18 **	2.70 ± 0.12 **
	Flibanserin	1.80 ± 0.11 **	1.74 ± 0.13 ** ^###^
	Iloperidone	1.50 ± 0.16	2.16 ± 0.15 ^##^
	Linagliptin	1.80 ± 0.12 **	2.34 ± 0.12 ^#^
	Nilotinib	1.32 ± 0.24 ^##^	1.80 ± 0.15 ** ^###^
	Paliperidone	1.44 ± 0.23 ^#^	2.4 ± 0.13 ^#^
	Vilazodone	1.56 ± 0.28 *	2.46 ± 0.08
HD1	DMSO	1.10 ± 0.10	1.50 ± 0.13
	Pridopidine	1.30 ± 0.19	1.80 ± 0.15
	Flibanserin	1.00 ± 0.19	2.40 ± 0.35 *** ^##^
	Iloperidone	1.40 ± 0.32	1.91 ± 0.23 *
	Linagliptin	1.60 ± 0.19 **	2.10 ± 0.20 **
	Nilotinib	1.38 ± 0.09	1.62 ± 0.03
	Paliperidone	1.80 ± 0.28 ** ^#^	2.20 ± 0.28 **
	Vilazodone	2.06 ± 0.20 *** ^##^	2.40 ± 0.28 *** ^##^
HD2	DMSO	1.08 ± 0.17 ^###^	1.87 ± 0.09 ^##^
	Pridopidine	1.56 ± 0.14 ***	1.56 ± 0.09 **
	Flibanserin	1.14 ± 0.16 ^##^	1.08 ± 0.09 *** ^###^
	Iloperidone	1.56 ± 0.15 ***	1.86 ± 0.06 ^##^
	Linagliptin	1.32 ± 0.13 * ^#^	1.38 ± 0.08 ***
	Nilotinib	1.20 ± 0.16 ^##^	2.14 ± 0.12 * ^###^
	Paliperidone	1.56 ± 0.12 ***	2.04 ± 0.07 ^###^
	Vilazodone	1.20 ± 0.22 ^##^	1.87 ± 0.12 ^##^

**Table 9 ijms-22-01293-t009:** Percentage CTR1, CTR2, HD1 and HD2 dead fibroblasts at 48 and 72 h after treatment with DMSO or 1 μM pridopidine, iloperidone, linagliptin, flibanserin, paliperidone, vilazodone or nilotinib. Percentages of dead cells are shown as mean ± standard deviation (SD). * *p* < 0.05 and ** *p* < 0.01 compared with DMSO treatment (Two-way ANOVA, followed by uncorrected Fisher’s LSD for multiple comparisons). ^#^
*p* < 0.05 and ^##^
*p* < 0.01 compared with pridopidine treatment (Two-way ANOVA, followed by uncorrected Fisher’s LSD for multiple comparisons).

Cell Line	Drug Treatment	% Dead Cells 48 h (Mean ± SD)	% Dead Cells 72 h (Mean ± SD)
CTR1	DMSO	7.43 ± 0.89	11.23 ± 2.02
	Pridopidine	6.62 ± 0.80	10.58 ± 1.91
	Flibanserin	9.24 ± 1.08	9.95 ± 0.88
	Iloperidone	12.81 ± 2.26 ** ^##^	13.24 ± 1.48
	Linagliptin	8.46 ± 1.00	9.01 ± 1.66
	Nilotinib	7.82 ± 0.93	10.81 ± 1.95
	Paliperidone	8.97 ± 1.05	12.17 ± 2.16
	Vilazodone	6.63 ± 0.80	9.18 ± 1.69
CTR2	DMSO	7.69 ± 0.91	10.58 ± 1.91
	Pridopidine	9.52 ± 1.11	9.97 ± 2.12
	Flibanserin	9.25 ± 1.08	14.32 ± 2.48
	Iloperidone	10.88 ± 1.25	11.87 ± 2.12
	Linagliptin	9.24 ± 1.08	11.06 ± 1.99
	Nilotinib	18.04 ± 3.00 ** ^##^	18.40 ± 3.92 ** ^##^
	Paliperidone	11.28 ± 1.29	11.86 ± 3.43
	Vilazodone	10.51 ± 1.21	10.84 ± 2.28
HD1	DMSO	14.27 ± 1.58	16.23 ± 2.75
	Pridopidine	12.35 ± 1.39	13.91 ± 2.42
	Flibanserin	15.48 ± 1.68	15.04 ± 1.06
	Iloperidone	11.57 ± 1.32	13.24 ± 2.33
	Linagliptin	10.27 ± 1.19 *	12.17 ± 2.16 *
	Nilotinib	14.00 ± 1.89	17.54 ± 2.71
	Paliperidone	7.78 ± 0.99 ** ^#^	10.64 ± 2.20 **
	Vilazodone	7.10 ± 0.54 ** ^##^	9.18 ± 0.36 ** ^#^
HD2	DMSO	13.06 ± 3.23	13.91 ± 2.42
	Pridopidine	10.51 ± 1.21	15.71 ± 2.68
	Flibanserin	20.3 ± 3.28 ** ^##^	21.19 ± 3.38 ** ^#^
	Iloperidone	10.52 ± 1.20	13.52 ± 2.37
	Linagliptin	12.18 ± 1.38	17.40 ± 2.91
	Nilotinib	13.24 ± 1.48	13.77 ± 0.41
	Paliperidone	10.51 ± 1.21	12.48 ± 2.21
	Vilazodone	13.24 ± 1.48	13.46 ± 2.36

## Data Availability

The data that support the findings of this study are available from the corresponding author upon reasonable request.

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
