# Peer review of "Known Drugs Identified by Structure-Based Virtual Screening Are Able to Bind Sigma-1 Receptor and Increase Growth of Huntington Disease Patient-Derived Cells"

_ijms, 2021, doi:10.3390/ijms22031293_

Round 1

Reviewer 1 Report

Battista et al., used molecular modeling of ligand-receptor interactions to screen FDA-approved drugs for their predicted ability to bind to the sigma-1 receptor (S1R). They found several with higher binding affinities than known ligands with clinical potential in the context of treating neurogenerative diseases. They confirmed predicted ligand-receptor interactions in vitro, monitoring binding of purified S1R with ligands via surface plasmon resonance. These results revealed comparable affinities with known ligands, albeit at a much lower affinity than in a cellular context. They obtained fibroblasts from two healthy people and two early-stage Huntington’s disease (HD) patients. Cellular growth and abundance were diminished in the fibroblast lines from HD patients compared to controls. Several ligands with high predicted binding affinity improved cellular proliferation, mimicking the action of pridopidine. As noted by the authors, these results could be strengthened by further evaluation of agonist properties in additional HD models (e.g., protection of neurons and/or synaptic connections) and by demonstrating that S1R inhibition or knockout blocks the action of identified agonists. Also, determining the Ki of the top ligands using a radioligand displacement assay with membrane bound S1R, along with known agonists for comparison, would help to confirm the binding affinity. In general, the paper was nicely written and is important in that it reveals new S1R ligands that are FDA-approved and provides a first-pass screening approach for predicting and confirming interactions between protein targets of clinical interest and FDA-approved drugs. This approach could be extended to other receptors and clinical indications.

Minor:

-If possible, provide info regarding the age and sex of the HD patients and healthy controls.

-Line 64: N,Nʹ dimethyl tryptamine, sphingosine, and myristic acid have been proposed to act as endogenous modulators of S1R.

-Line 67: Definition seems backwards. Agonists promote activity of a receptor, whereas antagonists block the action of an agonist.

-Line 70: change to: agonists increase IP3-dependent calcium flux at least with InsP3R3 and inhibit sodium . . .

-Lines 73-76: There is conflicting data and hypotheses regarding the action of agonists/antagonists on S1R oligomerization and function. In general, the ligand classes are defined in that the effect of agonists is blocked by antagonists. It is worth mentioning that there is evidence for differential effects of agonists and antagonists on oligomerization, but please be less conclusive unless this has been unambiguously resolved.

-Line 170 change should to may

-Line 172 change “an optimal” to “a”

-Line 193 make Authors lowercase and provide an explanation about RMSD values for better interpretation. Is this is Angstroms as indicated in the table 2 legend.

-Table 1: adjust the column widths. The formula column is too wide, making other columns less readable

-Table 3: for consistency and easy of comparison, use this order of identified agonists for the other tables and figures.

-Line 426: delete “determined a significant increase in either HD1 or HD2 fibroblast number at 72 h” to “significantly increased fibroblast number at 72 h in either HD1 or HD2 lines”

-Line 428: insert a after had

-Line 457: change cells to cell

-Line 472: change determines to causes

-Line 473: change and to and/or

-Line 474: delete “and neuromuscular problems” since this already fits with movement disorders.

-Line 482: change in to of

-Line 548: insert effects before for

-Line 586: change experiments to experiment

-Lines 615 and 619: change Nilotinib to Nilotinib’s

-Line 621: insert the before HD1

-Line 639: delete tests

-Line 642: change “as well as” to “and”. Also, change “time” to “time-“

-Line 646: change wider to larger

-Line 811: Add more details about how cells were counted. Was a device involved? What size of area was counted? How many regions per well? If manual, were people blind to the condition?

Author Response

Comments and Suggestions for Authors

Battista et al., used molecular modeling of ligand-receptor interactions to screen FDA-approved drugs for their predicted ability to bind to the sigma-1 receptor (S1R). They found several with higher binding affinities than known ligands with clinical potential in the context of treating neurogenerative diseases. They confirmed predicted ligand-receptor interactions in vitro, monitoring binding of purified S1R with ligands via surface plasmon resonance. These results revealed comparable affinities with known ligands, albeit at a much lower affinity than in a cellular context. They obtained fibroblasts from two healthy people and two early-stage Huntington’s disease (HD) patients. Cellular growth and abundance were diminished in the fibroblast lines from HD patients compared to controls. Several ligands with high predicted binding affinity improved cellular proliferation, mimicking the action of pridopidine.

As noted by the authors, these results could be strengthened by further evaluation of agonist properties in additional HD models (e.g., protection of neurons and/or synaptic connections) and by demonstrating that S1R inhibition or knockout blocks the action of identified agonists.

Also, determining the Ki of the top ligands using a radioligand displacement assay with membrane bound S1R, along with known agonists for comparison, would help to confirm the binding affinity.

In general, the paper was nicely written and is important in that it reveals new S1R ligands that are FDA-approved and provides a first-pass screening approach for predicting and confirming interactions between protein targets of clinical interest and FDA-approved drugs. This approach could be extended to other receptors and clinical indications.

Answer: We thank the reviewer for their appreciation of our work and their suggestions, which we have mentioned in the new version of the manuscript (section: “Discussion”) among the studies that we are going to perform on selected compounds in the future.

In particular, as far as the evaluation of agonist properties in additional HD cell models is concerned, our group possesses the expertise to generate induced pluripotent stem cells (iPSCs) and differentiate them into neurospheres (i.e., staminal neurons), which, in turn, can be differentiated into different types of neurons. These can be used, together with fibroblasts, to evaluate the ability of selected drugs to exert cell protection in the presence of S1R antagonists competing for the same binding site.

Additionally, we can take advantage of the equipment and expertise present in our department to determine the Ki of selected drugs by radioligand displacement assay with membrane bound S1R.

Please, note that all the changes introduced in the new version of the manuscript, to address the pints raised by this or other Reviewers, have been highlighted by blue font.

Minor:

  • If possible, provide info regarding the age and sex of the HD patients and healthy controls.

Answer: this information has been included in the new version of the manuscript (“Materials and Methods” section, “4.5. Skin biopsy and fibroblast isolation” subsection).

  • Line 64: N,Nʹ dimethyl tryptamine, sphingosine, and myristic acid have been proposed to act as endogenous modulators of S1R.

Answer: this information has been included in the new version of the manuscript

  • Line 67: Definition seems backwards. Agonists promote activity of a receptor, whereas antagonists block the action of an agonist.

Answer: we agree with the referee. In the new version of the manuscript, this paragraph has been corrected accordingly.

  • Line 70: change to: agonists increase IP3-dependent calcium flux at least with InsP3R3 and inhibit sodium . . .

Answer: In the new version of the manuscript, this sentence has been modified as suggested by the Reviewer.

  • Lines 73-76: There is conflicting data and hypotheses regarding the action of agonists/antagonists on S1R oligomerization and function. In general, the ligand classes are defined in that the effect of agonists is blocked by antagonists. It is worth mentioning that there is evidence for differential effects of agonists and antagonists on oligomerization, but please be less conclusive unless this has been unambiguously resolved.

Answer: In the new version of the manuscript, this sentence has been modified according to the Reviewer’s suggestion.

  • Line 170 change should to may

Answer: Done.

  • Line 172 change “an optimal” to “a”

Answer: Done.

  • Line 193 make Authors lowercase and provide an explanation about RMSD values for better interpretation. Is this is Angstroms as indicated in the table 2 legend.

Answer: Done.

  • Table 1: adjust the column widths. The formula column is too wide, making other columns less readable

Answer: Done.

  • Table 3: for consistency and easy of comparison, use this order of identified agonists for the other tables and figures.

Answer: As suggested by the Reviewer, we have changed the order of compounds in all the Tables in agreement with Table 3. However, since in Tables 7, 8 and 9, Pridopidine is used to calculate the statistical significance of the results obtained with the other compounds, we have decided to leave it before the others in those Tables and move it before the others in Table 3 and Figure 4 as well.

  • Line 426: delete “determined a significant increase in either HD1 or HD2 fibroblast number at 72 h” to “significantly increased fibroblast number at 72 h in either HD1 or HD2 lines”

Answer: Done.

  • Line 428: insert a after had

Answer: Done.

  • Line 457: change cells to cell

Answer: Done.

  • Line 472: change determines to causes

Answer: Done.

  • Line 473: change and to and/or

Answer: Done.

  • Line 474: delete “and neuromuscular problems” since this already fits with movement disorders.

Answer: Done.

  • Line 482: change in to of

Answer: We changed “consisting in” to “i.e.”, which we find more appropriate than “consisting of” in that context.

  • Line 548: insert effects before for

Answer: Done.

  • Line 586: change experiments to experiment

Answer: Done.

  • Lines 615 and 619: change Nilotinib to Nilotinib’s

Answer: Done.

  • Line 621: insert the before HD1

Answer: Done.

  • Line 639: delete tests

Answer: Done.

  • Line 642: change “as well as” to “and”. Also, change “time” to “time-“

Answer: Done.

  • Line 646: change wider to larger

Answer: Done.

  • Line 811: Add more details about how cells were counted. Was a device involved? What size of area was counted? How many regions per well? If manual, were people blind to the condition?

Answer: this information has been included in the new version of the manuscript (section: “Materials and Methods”; subsections: “4.6. Growth rate analysis in basal conditions” and “4.7. Growth rate analysis after drug treatment”).

Reviewer 2 Report

I’d like to start by congratulating the authors on telling a specific focused story, starting from generating a hypothesis through evaluating it computationally at the bench. This paper neatly outlines exactly the figures and findings needed to tell a single story. Huntington is an important indication representing an urgent unmet clinical need, it's great to see combined computational/experimental methods focused on repurposing in this indication.

Main comments

  1. What’s the mechanism of action of benefit for these drugs in fibroblasts, how do we know S1R is involved at all? For example: is Nilotinib benefiting cells by binding to S1R, or is it because of its inhibition of ABL, KIT, LCK, EPHA3, EPHA8, DDR2, PDGFRB, etc.  Iloperidone binds to 5HT2A, D2, and D3 with reported nanomolar affinity, and here the authors report 5uM binding to S1R. With such a difference, how can we know that the S1R activity is really contributing anything?  
  2. Why didn’t the authors filter for only Blood Brain Barrier crossing drugs from the start? If the goal is repurposing, a drug that doesn’t have physiologically relevant exposure levels in the striatum should not be considered in the first place. 
  3. The impact of this paper would be greatly increased if the authors would think deeper about how repurposing could actually work for these candidates that they propose. Each final candidate is commented on briefly in the discussion, but the thinking needs to steered more toward and expanded on how this could actually play out. For example, to pick on nilotinib again - it has a black box warning for heart complications. The HD population is very frail, how would repurposing such a toxic chemotherapeutic in such a cohort work? With a decreased dose to account for safety, will exposure in the striatum be high enough? Nilotinib is formulated as a tablet/capsule, will very ill HD patients with some swallow dysfunction be able to handle a tablet/capsule? The authors note that Nilotinib has been “investigated” for PD but fail to mention the cardiotox observed in a human trial: https://www.ncbi.nlm.nih.gov/pmc/articles/PMC5008228/#S1 For clarity, I am suggesting that the authors expand on all 6 final candidates, not just nilotinib. 

Minor comments:

  • The authors should comment on whether they would expect their fibroblast results to translate to induced neurons. For example, it’s known that neurons are more sensitive in culture, would they still expect to see the same growth? Neuronal cells also express different potential targets at different levels than fibroblasts. 
  • Figures 5, 6, 7: Show the dots on top of boxplots/barplots. Looks like the authors made these plots with Prism, here are instructions for adding the dots https://www.graphpad.com/support/faq/when-showing-a-box-and-whiskers-plot-that-also-shows-individual-data-points-how-to-put-the-points-behind-the-box-and-whiskers/ Or you could also plot with python matplotlib which also certainly will allow you to show the dots on top of boxplots and barplots https://stackoverflow.com/questions/29779079/adding-a-scatter-of-points-to-a-boxplot-using-matplotlib
  • “Results analysis. Python scripts were developed to parse …” Which version of python? Any particular python libraries used?

Author Response

Comments and Suggestions for Authors

I’d like to start by congratulating the authors on telling a specific focused story, starting from generating a hypothesis through evaluating it computationally at the bench. This paper neatly outlines exactly the figures and findings needed to tell a single story. Huntington is an important indication representing an urgent unmet clinical need, it's great to see combined computational/experimental methods focused on repurposing in this indication.

Answer: We thank the reviewer for their appreciation of our work.

Main comments

1. What’s the mechanism of action of benefit for these drugs in fibroblasts, how do we know S1R is involved at all? For example: is Nilotinib benefiting cells by binding to S1R, or is it because of its inhibition of ABL, KIT, LCK, EPHA3, EPHA8, DDR2, PDGFRB, etc.  Iloperidone binds to 5HT2A, D2, and D3 with reported nanomolar affinity, and here the authors report 5uM binding to S1R. With such a difference, how can we know that the S1R activity is really contributing anything?  

Answer: We thank the Referee for this comment. Like many known drugs, each of the drugs studied in our work is known, or likely, to interact with several receptors, some of which are quite diverse, as in the case of Nilotinib and Iloperidone mentioned by the Referee, and the resulting interaction may be responsible, at least in part, for their protective activity on HD fibroblasts. However, since S1R is the only receptor common to all of the studied drugs, and they have been selected among many (>1500) other compounds for their predicted ability to interact with S1R, we believe that it is unlikely, albeit possible, that the interaction with S1R does not contribute at all to their activity. This will have, of course, to be demonstrated and quantified, at least for those compounds that are going to be further investigated as HD therapeutics. To this end, as we have specified in the new version of the manuscript (section: “Discussion”), we plan to perform new experiments to measure the protective activity of selected drugs towards HD cell models in the presence of S1R antagonists.

2. Why didn’t the authors filter for only Blood Brain Barrier crossing drugs from the start? If the goal is repurposing, a drug that doesn’t have physiologically relevant exposure levels in the striatum should not be considered in the first place. 

Answer: The reason why we included all of the ZINC drugs in our analysis is that, although the ability to cross the blood-brain barrier is undoubtedly a desirable feature, compounds lacking of this feature are not prevented from being used as drugs, they just have to undergo different, although less convenient, administration routes. As an example, antisense oligonucleotides (ASOs) are currently in Phase III clinical trials for HD (see for example: https://en.hdbuzz.net/272) thanks to their ability to effectively lower huntingtin protein levels. However, since they cannot cross the blood-brain barrier, they are administered by injection into the spinal fluid, in spite of the fact that the injection may be painful and it must be performed in clinical settings, with frequency up to once a month. Given that HD is an unmet clinical need, the benefit of the therapeutic effect is estimated to outweigh the discomfort of the administration route. Therefore, we decided to include all the available drugs in the first stages of our analysis and postpone the evaluation of relative advantages vs. disadvantages to a later stage, once their relative S1R binding ability and protective activity on HD cell models had been experimentally assessed, to select the most suitable drugs for further characterization and studies in animal models. Interestingly, even if we had not set this condition as a pre-requisite for virtual screening, only Linagliptin has been reported not to cross intact blood-brain barrier. Conversely, Flibanserin, Iloperidone, Paliperidone and Vilazodone exert their therapeutic activity on brain receptors following oral administration; additionally, Nilotinib has been shown to be able to cross the blod-brain barrier following injection. Information about the blood-brain barrier crossing ability of the selected compounds has been included in the new version of the manuscript (section: “Discussion”).

3. The impact of this paper would be greatly increased if the authors would think deeper about how repurposing could actually work for these candidates that they propose. Each final candidate is commented on briefly in the discussion, but the thinking needs to steered more toward and expanded on how this could actually play out. For example, to pick on nilotinib again - it has a black box warning for heart complications. The HD population is very frail, how would repurposing such a toxic chemotherapeutic in such a cohort work? With a decreased dose to account for safety, will exposure in the striatum be high enough? Nilotinib is formulated as a tablet/capsule, will very ill HD patients with some swallow dysfunction be able to handle a tablet/capsule? The authors note that Nilotinib has been “investigated” for PD but fail to mention the cardiotox observed in a human trial: https://www.ncbi.nlm.nih.gov/pmc/articles/PMC5008228/#S1 For clarity, I am suggesting that the authors expand on all 6 final candidates, not just Nilotinib.

Answer: We thank again the reviewer again for their appreciation of our work and for raising important questions. As requested by the Reviewer, we have expanded the comments on each of the studied drugs in the new version of the manuscript (section: “Discussion”) to include information that may be relevant for a preliminary evaluation of repurposing feasibility. However, an accurate evaluation will require the acquisition of further information, which can only be provided by ad hoc designed experiments; these are also described more extensively in the new version of the manuscript (section: “Discussion”). Indeed, while approved drugs can, in principle, be prescribed off-label, in vivo studies in animal models, and eventually in the clinic, will have to be performed to acquire essential information, such as the optimal dosages at which each drug exerts the desired therapeutic effect with the lowest unwanted effects, the stage of the disease at which each drugs should be administered, etc.. As pointed out by the Reviewer, fragile advanced stages patients may require special attentions and ad hoc therapeutics; however, drugs that are inappropriate for these patients may turn out to effectively delay neurodegeneration in the early stages of the disease.

Please, note that all the changes introduced in the new version of the manuscript, to address the points raised by this or other Reviewers, have been highlighted by blue font.

Minor comments:

  • The authors should comment on whether they would expect their fibroblast results to translate to induced neurons. For example, it’s known that neurons are more sensitive in culture, would they still expect to see the same growth? Neuronal cells also express different potential targets at different levels than fibroblasts. 

Answer: as we have specified in the new version of the manuscript (section: “Discussion”), we believe that further experiments, including tests on iPSC-derived neurospheres and neurons, are required to evaluate the potential usefulness of the drugs studied in this work in HD therapy. As written in the Answer to Reviewer 1, our group possesses the expertise to generate induced pluripotent stem cells (iPSCs) and differentiate them into neurospheres (i.e., staminal neurons), which, in turn, can be differentiated into different types of neurons.

  • Figures 5, 6, 7: Show the dots on top of boxplots/barplots. Looks like the authors made these plots with Prism, here are instructions for adding the dots https://www.graphpad.com/support/faq/when-showing-a-box-and-whiskers-plot-that-also-shows-individual-data-points-how-to-put-the-points-behind-the-box-and-whiskers/ Or you could also plot with python matplotlib which also certainly will allow you to show the dots on top of boxplots and barplots https://stackoverflow.com/questions/29779079/adding-a-scatter-of-points-to-a-boxplot-using-matplotlib

Answer: In the new version of the manuscript, Figures 5, 6 and 7 have been modified as requested by the Reviewer.

  • “Results analysis. Python scripts were developed to parse …” Which version of python? Any particular python libraries used?

Answer: In the new version of the manuscript (section: Materials and Methods) we have now specified both the Python version and libraries used to write the scripts used to perform the analysis described in this work.

Reviewer 3 Report

In their article, authors use computational techniques complimented with experimental testing to investigate interactions of known drugs with sigma-1 receptor. Presently the title does not reflect the experimental portion of the work, which, in my opinion is even more important as a computational part. In my opinion, the title should be carefully revised.

Comparison of Vina and Chimera was not properly executed. Specifically, these programs report different metrics that could not directly be compared. The authors added ADT evaluation, but it would rather be biased towards Vina results. I would recommend revising analysis strategy to ensure that results obtained with different modeling tools are correctly communicated and compared.

Protein purification figure (Figure 3) is blurry. It would be helpful to add a figure of better quality. Same goes for Figure 4: the quality is too low, and it is barely readable. Authors should provide graphs of better quality.

It is unclear if authors found direct correlation with docking/screening scores and experimental data. Adding a discussion portion would help to build stronger connection between computation and experiment.

Overall, the manuscript could be published given that appropriate improvements and clarifications are done.

Author Response

Comments and Suggestions for Authors

In their article, authors use computational techniques complimented with experimental testing to investigate interactions of known drugs with sigma-1 receptor. Presently the title does not reflect the experimental portion of the work, which, in my opinion is even more important as a computational part. In my opinion, the title should be carefully revised.

Answer: We agree with the Reviewer. In the new version of the manuscript, the title has been modified to reflect the experimental, as well as computational part of the work.

Comparison of Vina and Chimera was not properly executed. Specifically, these programs report different metrics that could not directly be compared. The authors added ADT evaluation, but it would rather be biased towards Vina results. I would recommend revising analysis strategy to ensure that results obtained with different modeling tools are correctly communicated and compared.

Answer: We agree with the referee about the fact that the results produced by Vina and Chimera cannot be directly compared. As mentioned in the manuscript (sections: Materials and Methods and Results), we have used these two programs, and ADT, to perform different, sequential steps in our procedure: First, Vina was used to perform a virtual screening, calculate the interaction energy of each drug with the receptor and, based on these energies, provide a ranking of all the compounds. The 20 best ranking compounds were then docked again to the receptor binding site using ADT, which provided not only an estimate of interaction energy but also, and most importantly, clusters of drug conformations within the receptor binding sites. The reason to obtain these clusters is that, according to the authors of these programs, cluster size has higher correlation than predicted binding energy with experimentally determined affinity. Finally, Chimera was used to calculate interaction parameters, such as the number of favourable and unfavourable drug-receptor contacts. In other words, we have based our selection on visual inspection, performed with Chimera, of the lowest energy pose of the largest cluster produced by ADT, for each of the 20 best ranking compounds according to Vina. However, we reported the main results of all of the three programs in Table 2 to allow the reader to perform their own evaluation, and possibly use different criteria to select additional compounds to test experimentally. To facilitate Table 2 reading, in the new version of the manuscript, we have separated more clearly the results produced by each of the three programs.

Protein purification figure (Figure 3) is blurry. It would be helpful to add a figure of better quality. Same goes for Figure 4: the quality is too low, and it is barely readable. Authors should provide graphs of better quality.

Answer: we have replaced both Figure 3 and Figure 4 with higher quality images.

It is unclear if authors found direct correlation with docking/screening scores and experimental data. Adding a discussion portion would help to build stronger connection between computation and experiment.

Answer: as reported in the manuscript (section: Discussion) “there was no correlation between the predicted binding energy listed in Table 2 and actual binding affinities reported in Table 3”. This is not surprising, since, as also reported in the manuscript (section: Results) “the free binding energy predicted by docking methods has an accuracy of ~2-3 kcal/mol standard deviation [Huey R, Morris GM, Olson AJ, Goodsell DS. Software news and update a semiempirical free energy force field with charge-based desolvation. J Comput Chem. 2007 Apr 30;28(6):1145–52.], therefore ranking of poses based on this parameter alone is not reliable”. The reference is to a study performed by the Authors of these methods. In our case, the energy of the six selected compounds calculated by Vina are within 0.7 kcal/mol, well below the significance threshold reported by the authors of the methods. Even the energy of the best poses of the most populated clusters calculated by ADT for the same compounds are below 3 kcal/mol for all compounds except one (Linagliptin). This is why we relied on virtual screening and docking methods to perform an initial selection of a reasonable number of compounds to be visually inspected, but after this we performed interaction analysis to select the six compounds to be experimentally tested. For the same reason, as written in the manuscript (section: Discussion), we believe that the drug library screened in this work is likely to contain other sigma-1 receptor binders potentially endowed with protective activity in HD cell models, even if Vina ranked them, as well as several well-known sigma-1 receptor ligands, below the 20th position, as reported Table 2 of the manuscript.

Please, note that all the changes introduced in the new version of the manuscript, to address the points raised by this or other Reviewers, have been highlighted by blue font.